# Physiological and morphological correlates of blood parasite infection in urban and non-urban house sparrow populations

Coraline Bichet[1,2], François Brischoux[2], Cécile Ribout[2], Charline Parenteau[2], Alizée Meillère[2,3], Frédéric Angelier[2]*

1 Institute of Avian Research, Wilhelmshaven, Germany, 2 Centre d'Etudes Biologiques de Chizé, CNRS-Université de La Rochelle, UMR-7372, Villiers-en-Bois, France, 3 Centre for Integrative Ecology, School of Life & Environmental Sciences, Deakin University, Waurn Ponds, Victoria, Australia

* frederic.angelier@cebc.cnrs.fr

**Editor:** Érika Martins Braga, Universidade Federal de Minas Gerais, BRAZIL

**Data Availability Statement:** The data are available in: https://www.researchgate.net/publication/339102940_Physiological_and_morphological_

## Abstract

In the last decade, house sparrow populations have shown a general decline, especially in cities. Avian malaria has been recently suggested as one of the potential causes of this decline, and its detrimental effects could be exacerbated in urban habitats. It was initially thought that avian malaria parasites would not have large negative effects on wild birds because of their long co-evolution with their hosts. However, it is now well-documented that they can have detrimental effects at both the primo- and chronical infection stages. In this study, we examined avian malaria infection and its physiological and morphological consequences in four populations of wild house sparrows (2 urban and 2 rural). We did not find any relationship between the proportions of infected individuals and the urbanisation score calculated for our populations. However, we observed that the proportion of infected individuals increased during the course of the season, and that juveniles were less infected than adults. We did not detect a strong effect of malaria infection on physiological, morphological and condition indexes. Complex parasite dynamics and the presence of confounding factors could have masked the potential effects of infection. Thus, longitudinal and experimental studies are needed to understand the evolutionary ecology of this very common, but still poorly understood, wild bird parasite.

## Introduction

Avian haemosporidian parasites consist of three genera (*Plasmodium*, *Haemoproteus*, *Leucocytozoon*) transmitted by different insect vectors to birds [1], and have been extensively studied in the last few decades as a model in evolutionary ecology [2]. Avian haemosporidian parasites can be found in numerous species worldwide and their prevalence can reach very high percentages in some wild bird populations [3–9]. Due to a long co-evolution with their hosts, it was initially suggested that these parasites would not have large negative effects on wild birds [10]. On the other hand, these parasites had a huge detrimental effect on naive bird

correlates_of_blood_parasite_prevalence_in_differentially-urbanized_house_sparrow_populations.

**Funding:** This project was funded by the Agence Nationale de la Recherche (ANR-16-CE02-0004-01), the Centre National de la Recherche Scientifique, the Conseil Départemental des Deux-Sèvres, the CPER Econat and the Région Nouvelle-Aquitaine (projet MULTISTRESS). The funders had no role in study design, data collection and analysis, decision to publish, or preparation of the manuscript.

**Competing interests:** The authors have declared that no competing interests exist.

populations, especially following their introduction (e.g. the introduction of *Plasmodium relictum* and its vector *Culex quinquefasciatus* to the Hawaiian islands [11–13]).

The effects of avian malaria vary throughout the course of the infection. Following the primo infection, the host experiences an acute phase with a high level of parasitaemia [1], with important red blood cell destructions and tissue damages caused by the parasite development [14]. Infected individuals suffer from substantial costs, such as reduced activity [15], impaired growth [16], reduced immunity [17], poor body condition [18], anaemia [19, 20], and even death [21–23]. If the bird survives this acute phase, the infection enters into a chronic phase, characterized by a much lower parasitaemia alternates with cycles of parasitaemia recrudescence [24, 25]. This phase is also associated with significant costs, [20, 22], such as reduced breeding success [26] and performance [27–29].

In the last decade, urban house sparrow populations have declined in many cities [30–33], and interestingly, birds presenting higher avian malaria parasitaemia have recently been suggested as a possible cause of these population collapses [34]. Across avian species, there is mixed evidence for a difference in prevalence of haemosporidian parasites between urban and non-urban populations. While some studies found a higher prevalence in urban habitats [35–37], others found no difference [38] or even a lower prevalence [39–41]. Regarding house sparrows, one study found no difference in prevalence between urban and rural areas [7], while another study found a higher prevalence in non-urban birds, using both field and experimental data [42]. Nevertheless, the detrimental effect of malaria on survival and reproduction could be exacerbated in cities because of additional urban environmental constraints. Accordingly, several studies have shown that urbanisation is associated with multiple morphological and physiological changes in house sparrows. For example, urban sparrows are usually smaller, in poorer condition [43–45], and have a lower quality plumage [46] than rural ones. They also suffer from higher oxidative stress, and higher stress levels than rural sparrows [47–49]. These changes have been related to low food availability and quality [43, 44, 50, but see 51], high rates of pollution [47, 51], disturbance (traffic noise for example [52]), or even to a recent increase in predator pressure [53, 54]. However, the potential impact of malaria infection on morphological and physiological attributes still needs to be clarified in wild bird populations (see [45] for a study on house sparrows and see [20] for a study on red-winged blackbirds, *Agelaius phoeniceus*).

In this study, we examined avian malaria infection and its physiological and morphological correlates in four populations of wild house sparrows (2 urban and 2 rural). First, we investigated which factors could predict blood parasite infection. Particularly, we tested how malaria infection differs (1a) between populations (urban or rural) characterized by an urbanisation score, (1b) throughout the breeding season, and (1c) between juvenile and adult sparrows. Because previous studies comparing blood parasite prevalence between urban and non-urban populations produced mixed results, it was difficult to make predictions for the present study. Nevertheless, we expected that malaria infection would increase throughout the breeding season, since ambient temperature seems to be an important prevalence predictor, in temperate areas [55]. Similarly, we predicted that malaria infection would be higher in adults compared to juveniles because adults could have been infected during previous seasons [7]. Second, we explored if malaria infection status could predict several morphological (body size, body mass), physiological (stress hormones levels, haematocrit) and condition (fat and muscle scores, body condition) attributes. We tested whether infection status could (2a) affect the physiology, the morphology and the condition of house sparrows and (2b) account for the differences between urban and rural sparrows. In a previous paper using the same data set [44], we reported that urban house sparrows were in poorer body condition than rural sparrows,

and a previous study also suggested that malaria infection can reduce survival in adult and juvenile house sparrows [34]. Accordingly, we predicted that malaria infection would impair growth (i.e. reduced body size) and would be associated with a poor health status (i.e. low body condition, low haematocrit, high stress hormones levels). Avian malaria infection seems to be particularly detrimental following the primo infection [21, 56, 57], which is likely to occur at the juvenile stage when sparrows are especially sensitive to environmental and urban-related constraints [58, 59]. Therefore, we predicted that malaria infection would be especially detrimental in juveniles compared to adults. Finally, we also predicted that the detrimental influence of avian malaria would be exacerbated for urban sparrows because of the additional and cumulative constraints of the urban environment [43, 44, 47, 53].

## Material and methods

### Ethics statement

This work was conducted according to all institutional and national guidelines for animal care and use. The experimental protocols have been approved by the ethics committee of Poitou-Charentes, France (authorization number: CE2012-7). The permit for capture, sampling and banding was delivered by the 'Centre de Recherches sur la Biologie des Populations d'Oiseaux' (National Museum of Natural History, Paris) (permit number: 13794). The permits to sample public areas (CEBC and La Rochelle populations) were delivered by the 'Préfecture de la Charente-Maritime', the 'Préfecture des Deux-Sèvres' and the 'Centre d'Etudes Biologiques de Chizé' (hereafter CEBC).

### Study sites and sampling

We captured 113 house sparrows (68 adults and 45 juveniles) from four populations located in Western France (Table 1, Fig 1) using mist-nets during the 2013 breeding season (11$^{th}$ of May - 23$^{rd}$ of August), in the context of a previous study [44]. These populations were characterized by an urbanisation score determined in the previous study [44]. Two populations were located in medium-sized cities: La Rochelle (46°08'52.8"N, 1°09'12.7"W, 75,000 inhabitants, urbanisation score = 2.10) and Niort (46°18'46.4"N, 0°28'44.3"W, 58,000 inhabitants, urbanisation score = 1.61) (Fig 1, Table 1). The two other populations were located in rural habitats, either in a village (Villefollet, 46°07'37.7"N, 0°16'04.4"W, 200 inhabitants, urbanisation score = -1.21) or at a research station surrounded by a forest (CEBC, 46°08'50.5"N, 0°25'34.2"W, urbanisation score = -2.50) (Fig 1, Table 1).

Within 3 minutes after capture, we collected a blood sample from the brachial vein of each bird (150μl) to measure 'baseline corticosterone' levels [60]. The blood collected was also used to measure haematocrit and for molecular analyses (see below). We collected a second blood sample (150μl) after 30 minutes, to measure 'stress-induced corticosterone' levels [61]. Then, we banded birds with a numbered metal band, and measured their weight (hereafter 'body mass'), wing, and tarsus length (all individuals measured by one experimenter, A.M.). We determined fat and muscle scores as previously described [58, 62]. We calculated an index of body condition using the residuals of the linear regression of body mass vs. tarsus length ($F_{1,111}$ = 45.98, p < 0.001, $R^2$ = 0.49). We determined the age of the birds (adult or juvenile) based on plumage characteristics [63], and then released birds at their site of capture. In the case of an individual was recaptured, we immediately released it, to avoid any useless additional stress.

**Table 1. Urbanisation score, sample size (n), prevalence, physiological, morphological and condition values of the four studied house sparrow populations.**

| Population | Population type | Urbanisation score | N (adults/ juveniles) | Blood parasite prevalence | Haematocrit ± SD | Baseline corticosterone ± SD ($ng.ml^{-1}$) | Stress-induced corticosterone ± SD ($ng.ml^{-1}$) | Body mass ± SD (g) | Tarsus length ± SD (mm) | Wing length ± SD (mm) | Fat score ± SD | Muscle score ± SD |
|---|---|---|---|---|---|---|---|---|---|---|---|---|
| CEBC | Rural | -2.50 | 31 (18/13) | 0.35 | 0.46 ± 0.06 | 4.02 ± 3.68 | 35.57 ± 14.79 | 26.74 ± 2.34 | 18.71 ± 0.81 | 73.92 ± 3.54 | 1.42 ± 0.47 | 2.08 ± 0.47 |
| Villefollet | Rural | -1.21 | 28 (19/9) | 0.39 | 0.46 ± 0.06 | 3.15 ± 3.92 | 26.77 ± 12.36 | 27.23 ± 2.04 | 19.04 ± 0.88 | 74.71 ± 3.41 | 1.50 ± 0.45 | 2.30 ± 0.53 |
| Niort | Urban | 1.61 | 24 (15/9) | 0.42 | 0.47 ± 0.05 | 3.01 ± 3.12 | 20.70 ± 8.21 | 25.46 ± 2.47 | 18.30 ± 0.77 | 73.88 ± 4.29 | 1.77 ± 0.71 | 1.92 ± 0.41 |
| La Rochelle | Urban | 2.10 | 30 (16/14) | 0.40 | 0.45 ± 0.06 | 3.06 ± 3.23 | 29.76 ± 13.04 | 24.60 ± 1.92 | 17.92 ± 0.81 | 74.71 ± 3.41 | 1.85 ± 0.56 | 2.17 ± 0.53 |

SD: Standard deviation.

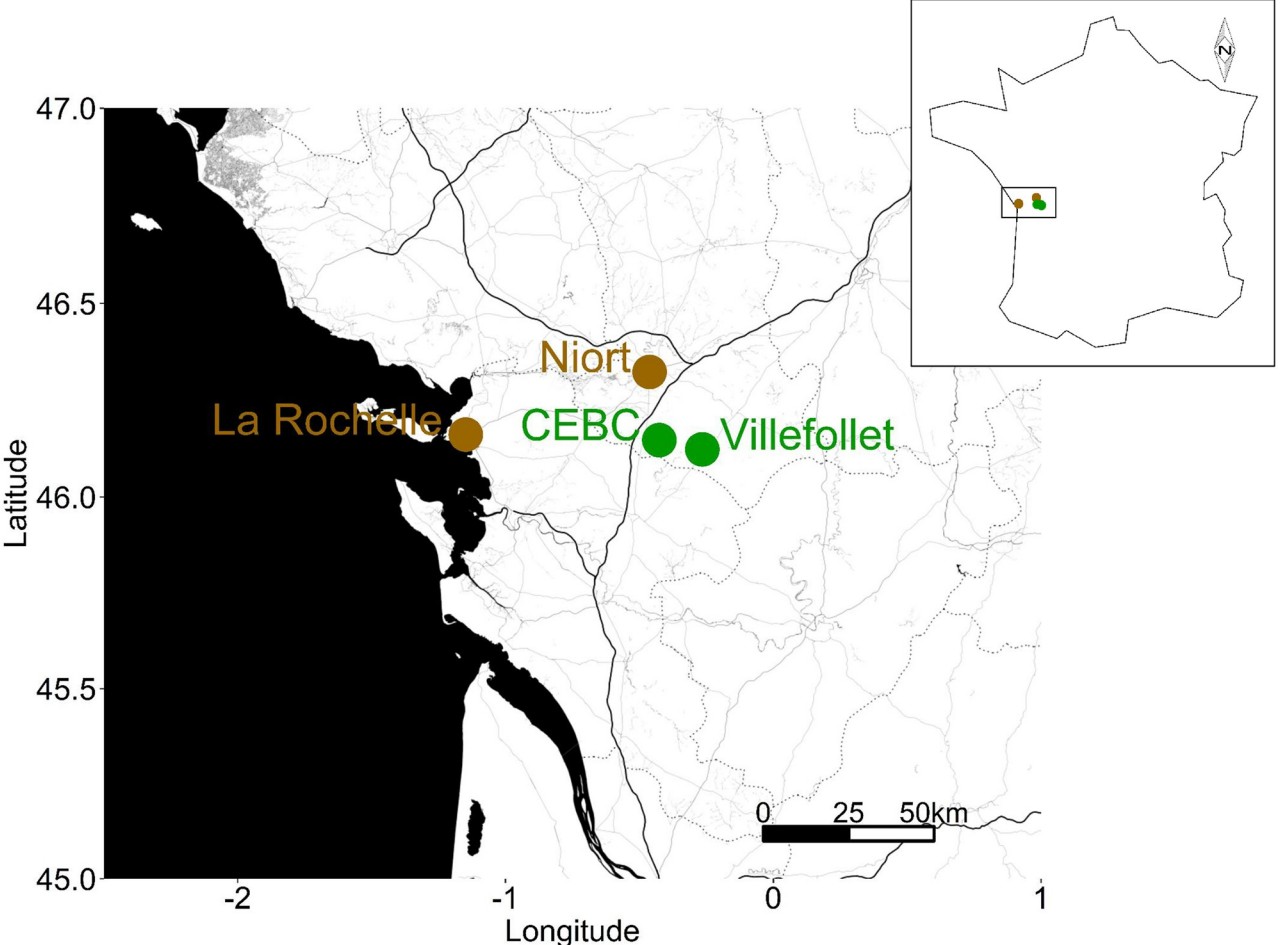

**Fig 1. Location of the four house sparrow populations sampled in this study.** Brown and green dots represent urban and rural sites, respectively.

## Physiological measurements

To measure haematocrit (determined by one experimenter, A.M.), we centrifuged 20µl of whole blood in a heparinized micro-capillary tube at 11,000 rpm for 3 minutes, and calculated the ratio between the volume of red blood cells and the total volume of whole blood (n = 113).

For each bird, we centrifuged blood samples (4,500 rpm, 7 min), separated plasma from red blood cells, transferred samples to separate tubes, and stored them at -20˚C until analyses. We measured plasma concentration of corticosterone, from the samples collected 3 and 30 minutes after capture, using a radio-immunoassay, in duplicate, according to the protocol developed by [64]. The minimum detectable corticosterone level was 0.28 ng.ml$^{-1}$, and the intra- and inter-assay coefficients of variation were 8.27% and 12.01%, respectively.

## Molecular analyses

We also stored red blood cells (approx. 75 µl) at -20˚C until analyses. We extracted DNA from red blood cells using the commercial kit NucleoSpin Blood (Macherey-Nagel Gmbh, Germany) and followed the provided instructions (n = 113). We used a Nanodrop (ND-1000) to standardize the concentration of the extracted DNA at 20ng.µl$^{-1}$, and used these standardized samples for molecular sexing and parasite screening analyses.

**Molecular sexing.** We determined the sex of adults visually (n = 68) [63], but determined the sex of juveniles (n = 45) using the molecular technique described in [65].

**Parasite screening.** We detected the presence of blood parasites using a nested PCR [66], which targets the cytochrome *b* gene of *Plasmodium*, *Leucocytozoon* and *Haemoproteus* in the extracted DNA (n = 113). We slightly modified the protocol described in [66] by using 40ng of total genomic DNA at 20ng. $\mu l^{-1}$, 0.8mM of each dNTP, 3.5mM of $MgCl_2$, 1.2$\mu$M of each primer and 0.625 units of Taq polymerase G2 Hot Start (Promega). The first PCR amplified a 580-bp-long fragment from the three blood parasite genera (primers HaemNFI/HaemNR3 [67, 68]). For the second PCRs, which amplified a 524-bp-long fragment, a first couple of primers (HaemF/HaemR2 [67, 68]) was used to detect indifferently *Plasmodium* or *Haemoproteus*, and a second couple of primers (HaemFL/HaemR2L [66]) to detect *Leucocytozoon*. This method is highly repeatable, with a minimum limit of detection of one infected blood cell per 100,000. We ran the products of the amplifications on a 1.8% agarose gel at 100V for 1h30, and visualized them with an ethidium bromide stain under ultraviolet light. We tested negative samples twice to minimize false negatives. We did not detect any positive blood sample for *Leucocytozoon*.

## Statistical analyses

To investigate which factors could predict blood parasite infection, we used a Generalized Linear Mixed Model (GLMM) with a logit link function and a variance given by a binomial distribution. The infection status of the individual (infected or uninfected) was the dependent variable and the age (categorical, adult or juvenile), the sex (categorical, male or female) of the bird, and the urbanisation score, were the explanatory variables. We also added capture date (standardized Julian date) as a covariate. The population was added as random intercept. To test if the effects of age and sex were consistent across the breeding season and the different populations, we also considered all first-order interactions.

To test if blood parasite infection could predict physiological, morphological and condition measures, we built nine different Linear Mixed Models (LMMs) using one of the physiological parameters (i.e. haematocrit, baseline corticosterone levels, and stress-induced corticosterone levels), morphological measures (i.e. body mass, wing, tarsus), or condition indexes (i.e. fat score, muscle score, and body condition) as the dependent variable. The models with fat and muscle scores as dependent variables were, however, two GLMs fitted with a Poisson distribution. Infection status (infected or uninfected) was added as the explanatory variable and the urbanisation score, age (categorical, adult or juvenile), and sex (categorical, male or female) were added as covariates. The population was added as random intercept. To test if the effect of the infection status was consistent across ages, sexes, urbanisation score and breeding season, we also considered all first-order interactions, as well as the second-order interaction between infection status, age, and sex.

In our dataset, there was collinearity between the urbanisation score, age, and capture date (Variance inflation factors $> 1/(1-R^2)$ [69]) due to the absence of juveniles and the absence of sampling in the La Rochelle population (with the higher urbanisation score) at the beginning of the season (from mid-May to mid-June). To limit this problem, we also ran LMMs similar to those presented above, but separating adults (n = 68) and juveniles (n = 45). One of the sampled juveniles was an outlier with a small body mass and short wing length. We ran our models with and without this individual and it did not modify our conclusions.

All models were run with R 3.6.0 [70] using the functions 'glmer' and 'lmer' implemented in the package 'lme4' [71] using restricted maximum likelihood estimates of the parameters, and by conducting a type III Wald $Chi^2$ tests using the package 'car' [72]. Non-significant

interactions were removed from the presented models in the Results, with the level of significance set to α = 0.05, to allow for a straightforward interpretation of the effect of single terms. Post-hoc tests for pairwise comparisons were conducted using the function 'contrast' from the R package 'lsmeans', after the least-square means were calculated using the function 'lsmeans' [73]. Parameter estimates are given as the mean ± 1 standard error (SE). The conditional coefficients of determination were calculated for all the models using the function 'r.squaredGLMM' implemented in the package 'MuMIn' [74].

## Results

### Blood parasite infection in house sparrows

The probability of being infected was not correlated with the urbanisation score (Table 2; Fig 2a). The probability of being infected increased during the breeding season, from May to August (Table 2, Fig 2b). Juvenile birds were less infected than adults (Table 2, Fig 2).

In juveniles, the probability of being infected increased during the breeding season (Table 2; Fig 2b), and male juveniles were more infected than female juveniles (Table 2).

In adults, the probability of being infected marginally increased during the breeding season (Table 2; Fig 2b) but did not differ between sexes (Table 2).

### Relationships between parasite infection and physiology

Infection status was not associated with any of the three physiological parameters investigated (haematocrit, baseline corticosterone levels, and stress-induced corticosterone levels, Table 3a, Fig 3a, 3b and 3c).

We also did not find an association between infection status and physiology in the models where adults and juveniles were analysed separately (Table 3a). We could mentioned, however, the marginal relationship between infection status and haematocrit in juveniles (infected juveniles had a slightly lower haematocrit than uninfected ones, Table 3a.1; Fig 3a).

### Relationships between parasite infection and morphology

Infection status did not predict any of the three morphological measures (body mass, tarsus length, wing length, Table 3b, Fig 3d, 3e and 3f). We detected an interaction between infection

**Table 2. Results from the GLMMs testing the relationship between age, sex, urbanisation score and capture date, and blood parasite infection status for all house sparrows (n = 113), adults (n = 68) and juvenile (n = 45) without non-significant interaction terms.** Significant effects (p < 0.05) are highlighted in bold. Conditional $R^2$ provided the proportion of explained variance.

| Dependent variable | Infection status | | | | | | | | |
|---|---|---|---|---|---|---|---|---|---|
| | All birds (n = 113) | | | Adults (n = 68) | | | Juveniles (n = 45) | | |
| **Parameter** | **Estimate ± SE** | **z-value** | **p-value** | **Estimate ± SE** | **z-value** | **p-value** | **Estimate ± SE** | **z-value** | **p-value** |
| Intercept | -0.349 ± 0.340 | -1.025 | 0.305 | -0.185 ± 0.365 | -0.508 | 0.612 | **-3.437 ± 1.461** | **-2.999** | **0.003** |
| Age (juvenile) | **-1.288 ± 0.481** | **-2.679** | **0.007** | - | - | - | - | - | - |
| Sex (male) | 0.720 ± 0.421 | 1.711 | 0.087 | 0.303 ± 0.503 | 0.603 | 0.546 | **1.986 ± 0.872** | **2.278** | **0.023** |
| Urbanisation score | $4.888.10^{-3}$ ± 0.11 | 0.044 | 0.965 | -0.017 ± 0.136 | -0.128 | 0.898 | 0.100 ± 0.205 | 0.468 | 0.640 |
| Capture date | **0.622 ± 0.248** | **2.508** | **0.012** | 0.449 ± 0.261 | 1.722 | 0.085 | **2.085 ± 0.980** | **2.127** | **0.033** |
| Random Population (variance ± SD) | 0.000 ± 0.000 | | | 0.000 ± 0.000 | | | 0.000 ± 0.000 | | |
| Conditional $R^2$ | 0.140 | | | 0.063 | | | 0.421 | | |

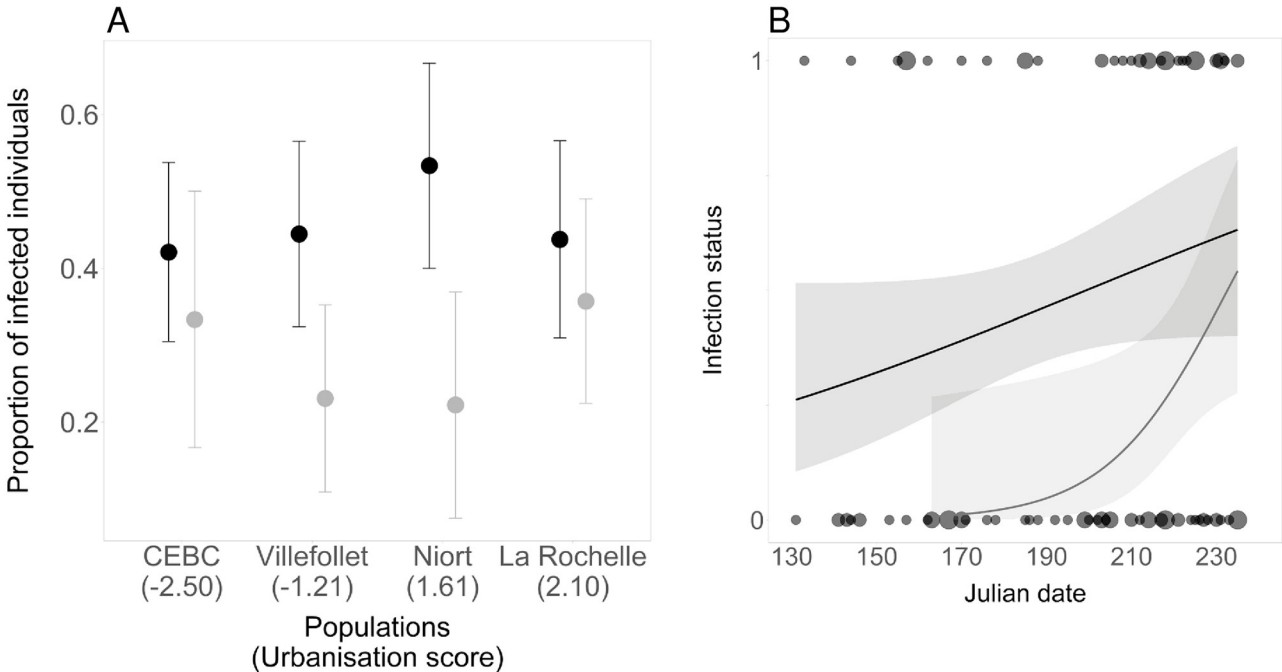

**Fig 2.** (A) Proportion of infected individuals in relation to population and urbanisation score, and (B) infection status (0 = uninfected, 1 = infected) in relation to capture date, in adults and juvenile house sparrows. (A) Dots with error bars indicate the means (± SE). Black dots correspond to the adults and grey dots to the juveniles. (B) Dots represent the infection status of the individuals. The black and grey solid lines with standard errors (grey areas) represent the model predictions for adults and juveniles, respectively.

status, age and sex for the body mass (Table 3b.1) and for the tarsus length (Table 3b.2), indicating that the influence of infection status on body mass and tarsus length differed between juveniles and adults (Fig 3d and 3e) and between males and females. However, post-hoc comparisons indicated that body mass and tarsus length did not significantly differ between infected or non-infected juveniles and adults, or males and females (all p > 0.300).

When adults were analysed separately, the infection status was not associated with body mass, tarsus length, nor wing length (Table 3b). We observed an interaction between infection status and sex for the body mass (Table 3b.1, S1a Fig) and for the tarsus length (Table 3b.2, S1b Fig). For instance, infected females seemed to have a lower body mass than non-infected females, while infected males seemed to have a higher body mass than non-infected males (S1a Fig), although post-hoc comparisons were not statistically significant (all p-values > 0.150).

When juveniles were analysed separately, we found an interaction between infection status and capture date for tarsus length (Table 3b.2), probably because both juvenile size and malaria infection increased throughout the breeding season.

## Relationships between parasite infection and condition

Infection status was not associated with sparrow condition (body condition, fat, and muscle scores, Table 3c). We only found an interaction between infection status and age for the body condition index (Table 3c.3), indicating that the influence of infection status on body condition differed between adults and juveniles (Fig 3g). This was confirmed by additional analyses: when adults were analysed separately, infection status was not associated with body condition (Table 3c.3). However, in adults, we observed an interaction between infection status and urbanisation score for the body condition index (Table 3c.3, S2 Fig). When juveniles were

**Table 3. Results from the LMMs testing the relationship between infection status, age, sex, and urbanisation score, and (A) physiological parameters, (B) morphological measurements and (C) condition indices for all house sparrows (n = 113), adults (n = 68) and juvenile (n = 45) without non-significant interaction terms.**

| A. Physiological parameters | | | | | | | | | |
|---|---|---|---|---|---|---|---|---|---|
| Dependent variable | **1. Haematocrit** | | | | | | | | |
| | All birds (n = 113) | | | Adults (n = 68) | | | Juveniles (n = 45) | | |
| Parameter | Estimate ± SE | t-value | p-value | Estimate ± SE | t-value | p-value | Estimate ± SE | t-value | p-value |
| Intercept | **0.473 ± 0.009** | **48.209** | **<0.001** | **0.469 ± 0.013** | **36.724** | **<0.001** | **0.433 ± 0.012** | **34.772** | **<0.001** |
| Infection status (infected) | $-6.339.10^{-3} \pm 1.043.10^{-2}$ | -0.608 | 0.545 | $1.780.10^{-3} \pm 1.324.10^{-2}$ | 0.134 | 0.893 | $-3.177.10^{-2} \pm 1.759.10^{-2}$ | -1.81 | 0.078 |
| Age (juvenile) | **$-2.836.10^{-2} \pm 1.144.10^{-2}$** | **-2.479** | **0.015** | - | - | - | - | - | - |
| Sex (male) | $9.704.10^{-4} \pm 9.911.10^{-3}$ | 0.098 | 0.922 | $-1.194.10^{-3} \pm 1.304.10^{-2}$ | -0.092 | 0.927 | $1.409.10^{-2} \pm 1.613.10^{-2}$ | 0.873 | 0.388 |
| Urbanisation score | $2.941.10^{-3} \pm 3.138.10^{-3}$ | 0.937 | 0.441 | $2.805.10^{-3} \pm 4.749.10^{-3}$ | 0.591 | 0.614 | $2.713.10^{-3} \pm 3.932.10^{-3}$ | 0.690 | 0.494 |
| Capture date | **$-1.979.10^{-2} \pm 5.865.10^{-3}$** | **-3.374** | **0.001** | **$-2.477.10^{-2} \pm 6.816.10^{-3}$** | **-3.634** | **<0.001** | $2.355.10^{-3} \pm 1.246.10^{-2}$ | 0.189 | 0.851 |
| Random Population (variance ± SD) | $4.411.10^{-5} \pm 6.642.10^{-3}$ | | | $1.513.10^{-4} \pm 1.230.10^{-2}$ | | | $0.000 \pm 0.000$ | | |
| Conditional $R^2$ | 0.246 | | | 0.210 | | | 0.079 | | |
| Dependent variable | **2. Baseline corticosterone** | | | | | | | | |
| | All birds (n = 113) | | | Adults (n = 68) | | | Juveniles (n = 45) | | |
| Parameter | Estimate ± SE | t-value | p-value | Estimate ± SE | t-value | p-value | Estimate ± SE | t-value | p-value |
| Intercept | **4.461 ± 0.554** | **8.046** | **<0.001** | **4.439 ± 0.799** | **5.556** | **<0.001** | **1.947 ± 0.362** | **5.368** | **<0.001** |
| Infection status (infected) | -0.142 ± 0.618 | -0.230 | 0.819 | $6.545.10^{-2} \pm 0.933$ | -0.070 | 0.944 | -0.869 ± 0.512 | -1.696 | 0.098 |
| Age (juvenile) | **-2.156 ± 0.679** | **-3.177** | **0.002** | - | - | - | - | - | - |
| Sex (male) | -0.849 ± 0.596 | -1.425 | 0.157 | -1.117 ± 0.921 | -1.214 | 0.230 | $-3.699.10^{-2} \pm 0.469$ | -0.079 | 0.938 |
| Urbanisation score | -0.054 ± 0.155 | -0.346 | 0.730 | $5.577.10^{-2} \pm 0.268$ | 0.208 | 0.836 | $9.483.10^{-2} \pm 0.115$ | 0.828 | 0.413 |
| Capture date | **-0.770 ± 0.361** | **-2.135** | **0.035** | -0.947 ± 0.489 | -1.937 | 0.057 | -0.111 ± 0.363 | -0.305 | 0.762 |
| Urbanisation score:capture date | **0.386 ± 0.159** | **2.431** | **0.017** | **0.516 ± 0.255** | **2.022** | **0.048** | - | - | - |
| Random Population (variance ± SD) | $0.000 \pm 0.000$ | | | $0.000 \pm 0.000$ | | | $0.000 \pm 0.000$ | | |
| Conditional $R^2$ | 0.282 | | | 0.179 | | | 0.098 | | |
| Dependent variable | **3. Stress-induced corticosterone** | | | | | | | | |
| | All birds (n = 113) | | | Adults (n = 68) | | | Juveniles (n = 45) | | |
| Parameter | Estimate ± SE | t-value | p-value | Estimate ± SE | t-value | p-value | Estimate ± SE | t-value | p-value |
| Intercept | **33.669 ± 3.230** | **10.424** | **0.002** | **34.303 ± 3.283** | **10.446** | **<0.001** | **19.868 ± 3.585** | **5.542** | **0.003** |
| Infection status (infected) | -1.486 ± 1.830 | -0.812 | 0.419 | -2.081 ± 2.499 | -0.833 | 0.408 | -1.185 ± 2.563 | -0.462 | 0.647 |
| Age (juvenile) | **-8.334 ± 2.028** | **-4.109** | **<0.001** | - | - | - | - | - | - |
| Sex (male) | **-4.256 ± 1.765** | **-2.412** | **0.018** | **-6.056 ± 2.488** | **-2.434** | **0.018** | 2.553 ± 3.053 | 0.836 | 0.408 |
| Urbanisation score | **-0.963 ± 1.520** | **-0.633** | **0.591** | -0.372 ± 1.489 | -0.250 | 0.824 | -0.812 ± 1.491 | -0.544 | 0.640 |
| Capture date | **-5.452 ± 1.083** | **-5.034** | **<0.001** | **-6.246 ± 1.323** | **-4.722** | **<0.001** | 2.031 ± 2.977 | 0.682 | 0.499 |
| Urbanisation score:capture date | **1.202 ± 0.482** | **2.491** | **0.014** | **1.762 ± 0.699** | **2.522** | **0.014** | - | - | - |
| Sex (male):capture date | - | - | - | - | - | - | **-7.527 ± 3.492** | **-2.156** | **0.038** |
| Random Population (variance ± SD) | 30.920 ± 5.561 | | | 24.940 ± 4.994 | | | 27.931 ± 5.285 | | |
| Conditional $R^2$ | 0.607 | | | 0.536 | | | 0.476 | | |
| B. Morphological parameters | | | | | | | | | |
| Dependent variable | **1. Body mass** | | | | | | | | |
| | All birds (n = 113) | | | Adults (n = 68) | | | Juveniles (n = 45) | | |
| Parameter | Estimate ± SE | t-value | p-value | Estimate ± SE | t-value | p-value | Estimate ± SE | t-value | p-value |
| Intercept | 26.856 ± 0.496 | 54.146 | <0.001 | **26.920 ± 0.414** | **65.047** | **<0.001** | **23.860 ± 0.635** | **37.591** | **<0.001** |

*(Continued)*

**Table 3.** (*Continued*)

| | Estimate ± SE | t-value | p-value | Estimate ± SE | t-value | p-value | Estimate ± SE | t-value | p-value |
|---|---|---|---|---|---|---|---|---|---|
| Infection status (infected) | -0.334 ± 0.631 | -0.529 | 0.598 | -0.521 ± 0.504 | -1.034 | 0.305 | 0.253 ± 0.852 | 0.297 | 0.768 |
| Age (juvenile) | **-3.147 ± 0.650** | **-4.842** | **<0.001** | - | - | - | - | - | - |
| Sex (male) | -0.480 ± 0.582 | -0.825 | 0.411 | -0.511 ± 0.467 | -1.094 | 0.278 | 0.125 ± 0.784 | 0.159 | 0.874 |
| Urbanisation score | **-0.636 ± 0.190** | **-3.346** | **0.037** | -0.439 ± 0.159 | -2.767 | 0.106 | -0.461 ± 0.216 | -2.129 | 0.172 |
| Capture date | **-0.486 ± 0.222** | **-2.184** | **0.031** | **-0.487 ± 0.181** | **-2.690** | **0.009** | 0.991 ± 0.607 | 1.633 | 0.110 |
| Infection status (infected): age (juvenile) | 1.561 ± 1.160 | 1.346 | 0.181 | - | - | - | - | - | - |
| Infection status (infected):sex (male) | 1.394 ± 0.862 | 1.617 | 0.109 | **1.475 ± 0.693** | **2.129** | **0.037** | - | - | - |
| Sex (male): age (juvenile) | 1.286 ± 0.921 | 1.396 | 0.166 | - | - | - | - | - | - |
| Age (juvenile):capture date | **1.635 ± 0.501** | **3.267** | **0.001** | - | - | - | - | - | - |
| Urbanisation score:sex (male) | **0.384 ± 0.176** | **2.178** | **0.032** | - | - | - | - | - | - |
| Infection status (infected):age (juvenile):sex (male) | **-3.311 ± 1.531** | **-2.162** | **0.033** | - | - | - | - | - | - |
| Random Population (variance ± SD) | 0.289 ± 0.538 | | | 0.241 ± 0.491 | | | 0.154 ± 0.393 | | |
| Conditional R$^2$ | 0.511 | | | 0.466 | | | 0.172 | | |
| Dependent variable | **2. Tarsus length** | | | | | | | | |
| | All birds (n = 113) | | | Adults (n = 68) | | | Juveniles (n = 45) | | |
| Parameter | Estimate ± SE | t-value | p-value | Estimate ± SE | t-value | p-value | Estimate ± SE | t-value | p-value |
| Intercept | **18.778 ± 0.259** | **72.594** | **<0.001** | **18.825 ± 0.273** | **68.952** | **<0.001** | **17.878 ± 0.148** | **120.721** | **<0.001** |
| Infection status (infected) | -0.466 ± 0.250 | -1.862 | 0.066 | -0.453 ± 0.276 | -1.642 | 0.106 | -0.795 ± 0.480 | -1.657 | 0.106 |
| Age (juvenile) | **-1.105 ± 0.259** | **-4.268** | **<0.001** | - | - | - | - | - | - |
| Sex (male) | -0.295 ± 0.233 | -1.267 | 0.208 | -0.335 ± 0.256 | -1.306 | 0.197 | 0.192 ± 0.191 | 1.005 | 0.321 |
| Urbanisation score | -0.137 ± 0.109 | -1.254 | 0.337 | -0.174 ± 0.118 | -1.476 | 0.275 | -0.081 ± 0.047 | -1.742 | 0.089 |
| Capture date | **-0.255 ± 0.092** | **-2.784** | **0.006** | **-0.285 ± 0.100** | **-2.861** | **0.006** | -0.081 ± 0.152 | -0.531 | 0.598 |
| Infection status (infected): age (juvenile) | **1.206 ± 0.461** | **2.614** | **0.010** | - | - | - | - | - | - |
| Infection status (infected):sex (male) | **0.769 ± 0.343** | **2.240** | **0.027** | **0.774 ± 0.380** | **2.041** | **0.046** | - | - | - |
| Infection status (infected):capture date | - | - | - | - | - | - | **1.527 ± 0.518** | **2.950** | **0.005** |
| Sex (male): age (juvenile) | **0.745 ± 0.370** | **2.016** | **0.046** | - | - | - | - | - | - |
| Age (juvenile):capture date | **0.420 ± 0.200** | **2.104** | **0.038** | - | - | - | - | - | - |
| Urbanisation score:sex (male) | - | - | - | - | - | - | - | - | - |
| Urbanisation score:capture date | **0.079 ± 0.038** | **2.092** | **0.039** | - | - | - | - | - | - |
| Infection status (infected):age (juvenile):sex (male) | **-1.290 ± 0.611** | **-2.112** | **0.037** | - | - | - | - | - | - |
| Random Population (variance ± SD) | 0.156 ± 0.395 | | | 0.165 ± 0.406 | | | 0.000 ± 0.000 | | |
| Conditional R$^2$ | 0.515 | | | 0.454 | | | 0.352 | | |
| Dependent variable | **3. Wing length** | | | | | | | | |
| | All birds (n = 113) | | | Adults (n = 68) | | | Juveniles (n = 45) | | |
| Parameter | Estimate ± SE | t-value | p-value | Estimate ± SE | t-value | p-value | Estimate ± SE | t-value | p-value |
| Intercept | **74.375 ± 0.399** | **186.237** | **<0.001** | **74.168 ± 0.367** | **201.898** | **<0.001** | **69.810 ± 0.746** | **93.544** | **<0.001** |
| Infection status (infected) | 0.622 ± 0.451 | 1.379 | 0.171 | 0.559 ± 0.434 | 1.288 | 0.203 | 1.011 ± 1.054 | 0.959 | 0.343 |
| Age (juvenile) | **-4.980 ± 0.492** | **-10.112** | **<0.001** | - | - | - | - | - | - |
| Sex (male) | **2.400 ± 0.428** | **5.600** | **<0.001** | **2.864 ± 0.423** | **6.759** | **<0.001** | 1.548 ± 0.967 | 1.602 | 0.117 |
| Urbanisation score | -0.113 ± 0.113 | -0.100 | 0.920 | -0.049 ± 0.115 | -0.428 | 0.670 | $2.367.10^{-3} \pm 0.236$ | -0.010 | 0.992 |
| Capture date | -0.334 ± 0.252 | -1.327 | 0.187 | -0.308 ± 0.221 | -1.394 | 0.168 | -0.541 ± 0.747 | -0.724 | 0.473 |

(*Continued*)

**Table 3.** (Continued)

| | All birds (n = 113) | | | Adults (n = 68) | | | Juveniles (n = 45) | | |
|---|---|---|---|---|---|---|---|---|---|
| Random Population (variance ± SD) | 0.000 ± 0.000 | | | 0.000 ± 0.000 | | | 0.000 ± 0.000 | | |
| Conditional R² | 0.648 | | | 0.439 | | | 0.128 | | |
| **C. Condition indexes** | | | | | | | | | |
| Dependent variable | **1. Fat score** | | | | | | | | |
| | All birds (n = 113) | | | Adults (n = 68) | | | Juveniles (n = 45) | | |
| Parameter | Estimate ± SE | t-value | p-value | Estimate ± SE | t-value | p-value | Estimate ± SE | t-value | p-value |
| Intercept | **1.073 ± 0.103** | **10.444** | **<0.001** | **1.155 ± 0.121** | **9.510** | **<0.001** | **1.261 ± 0.139** | **9.029** | **<0.001** |
| Infection status (infected) | -0.055 ± 0.115 | -0.483 | 0.629 | -0.101 ± 0.150 | -0.676 | 0.499 | $-3.837.10^{-2} \pm 0.186$ | -0.206 | 0.837 |
| Age (juvenile) | **0.261 ± 0.122** | **2.140** | **0.032** | - | - | - | - | - | - |
| Sex (male) | 0.016 ± 0.109 | 0.148 | 0.882 | $9.351.10^{-2} \pm 0.146$ | -0.642 | 0.521 | 0.174 ± 0.172 | 1.008 | 0.314 |
| Urbanisation score | 0.047 ± 0.029 | 1.625 | 0.104 | $8.081.10^{-3} \pm 3.955.10^{-2}$ | 0.204 | 0.838 | $\mathbf{9.716.10^{-2} \pm 4.266.10^{-2}}$ | **2.278** | **0.023** |
| Capture date | 0.068 ± 0.067 | 1.023 | 0.306 | $9.309.10^{-2} \pm 7.646.10^{-2}$ | 1.217 | 0.223 | $2.872.10^{-2} \pm 0.139$ | 0.207 | 0.836 |
| Random Population (variance ± SD) | 0.000 ± 0.000 | | | 0.000 ± 0.000 | | | 0.000 ± 0.000 | | |
| Conditional R² | 0.137 | | | 0.043 | | | 0.138 | | |
| Dependent variable | **2. Muscle score** | | | | | | | | |
| | All birds (n = 113) | | | Adults (n = 68) | | | Juveniles (n = 45) | | |
| Parameter | Estimate ± SE | t-value | p-value | Estimate ± SE | t-value | p-value | Estimate ± SE | t-value | p-value |
| Intercept | **1.469 ± 0.088** | **16.761** | **<0.001** | **1.403 ± 0.103** | **13.577** | **<0.001** | **1.399 ± 0.133** | **10.547** | **<0.001** |
| Infection status (infected) | -0.064 ± 0.099 | -0.643 | 0.521 | $-6.686.10^{-2} \pm 0.119$ | -0.563 | 0.573 | $-4.096.10^{-2} \pm 0.189$ | -0.217 | 0.829 |
| Age (juvenile) | -0.097 ± 0.110 | -0.884 | 0.376 | - | - | - | - | - | - |
| Sex (male) | 0.071 ± 0.094 | 0.761 | 0.447 | 0.175 ± 0.116 | 1.510 | 0.131 | $9.867.10^{-2} \pm 0.173$ | -0.570 | 0.569 |
| Urbanisation score | $3.268.10^{-3} \pm 2.485.10^{-2}$ | 0.132 | 0.895 | $2.781.10^{-3} \pm 3.137.10^{-2}$ | -0.089 | 0.929 | $-5.04.10^{-3} \pm 4.22.10^{-2}$ | -0.119 | 0.905 |
| Capture date | -0.050 ± 0.054 | -0.925 | 0.355 | $-7.359.10^{-2} \pm 5.986.10^{-2}$ | -1.229 | 0.219 | $4.273.10^{-2} \pm 0.135$ | 0.317 | 0.751 |
| Random Population (variance ± SD) | 0.000 ± 0.000 | | | 0.000 ± 0.000 | | | 0.000 ± 0.000 | | |
| Conditional R² | 0.041 | | | 0.075 | | | 0.018 | | |
| Dependent variable | **3. Body condition** | | | | | | | | |
| | All birds (n = 113) | | | Adults (n = 68) | | | Juveniles (n = 45) | | |
| Parameter | Estimate ± SE | t-value | p-value | Estimate ± SE | t-value | p-value | Estimate ± SE | t-value | p-value |
| Intercept | -0.032 ± 0.140 | -0.225 | 0.826 | $-5.070.10^{-2} \pm 0.185$ | -0.274 | 0.794 | $-8.445.10^{-2} \pm 0.126$ | -0.668 | 0.508 |
| Infection status (infected) | -0.124 ± 0.156 | -0.794 | 0.429 | $-9.444.10^{-2} \pm 0.170$ | -0.554 | 0.582 | **0.407 ± 0.178** | **2.281** | **0.028** |
| Age (juvenile) | -0.036 ± 0.167 | -0.215 | 0.830 | - | - | - | - | - | - |
| Sex (male) | 0.067 ± 0.123 | 0.546 | 0.586 | $5.616.10^{-2} \pm 0.170$ | 0.329 | 0.743 | 0.162 ± 0.164 | 0.986 | 0.330 |
| Urbanisation score | -0.032 ± 0.048 | -0.662 | 0.573 | -0.153 ± 0.085 | -1.812 | 0.157 | $3.098.10^{-2} \pm 3.991.10^{-2}$ | 0.776 | 0.442 |
| Capture date | **-0.155 ± 0.073** | **-2.130** | **0.036** | -0.146 ± 0.088 | -1.657 | 0.103 | -0.231 ± 0.126 | -1.828 | 0.075 |
| Infection status (infected):age (juvenile) | **0.536 ± 0.260** | **2.069** | **0.041** | - | - | - | - | - | - |
| Infection status (infected): urbanisation score | - | - | - | **0.193 ± 0.088** | **2.199** | **0.032** | - | - | - |
| Random Population (variance ± SD) | 0.018 ± 0.135 | | | $5.279.10^{-2} \pm 0.298$ | | | 0.000 ± 0.000 | | |
| Conditional R² | 0.140 | | | 0.236 | | | 0.196 | | |

Significant effects (p < 0.05) are highlighted in bold. Conditional $R^2$ provided the proportion of explained variance. '-' means a parameter was not fitted to the model and ':' represents interactions.

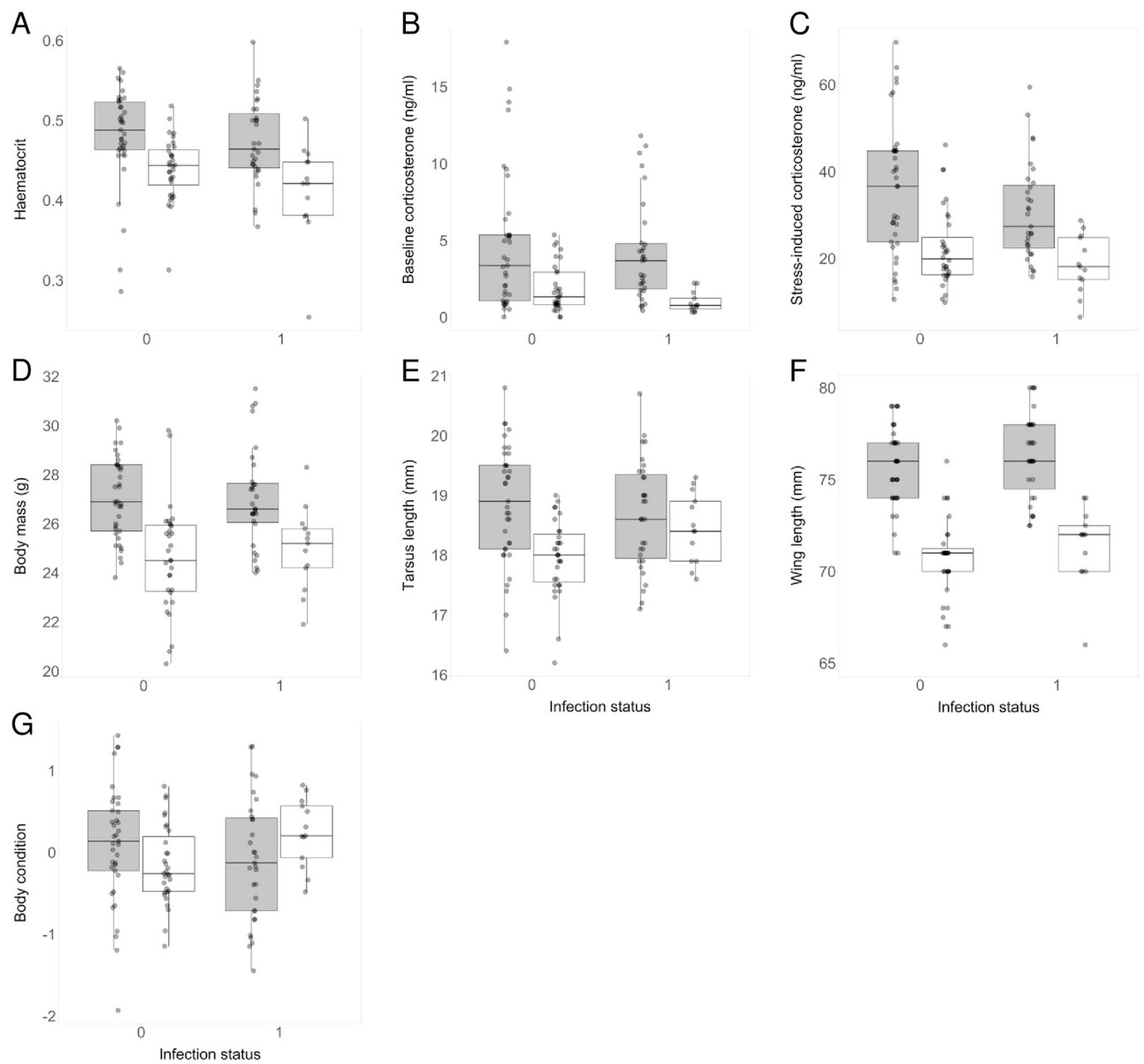

**Fig 3. (A) Haematocrit, (B) baseline corticosterone level, (C) stress-induced corticosterone level, (D) body mass, (E) tarsus length, (F) wing length and (G) body condition in relation to infection status (0 = uninfected, 1 = infected) and age.** Dots represent the raw data. Grey boxplots represent adults and white boxplots represent juveniles.

analysed separately, we found an association between infection status and body condition (Table 3c.3), indicating that infected juveniles had a higher body condition than uninfected juveniles (mean ± SE of body condition for infected juveniles = 0.24 ± 0.12, mean of body condition ± SE for uninfected juveniles = -0.12 ± 0.09; Fig 3g).

## Discussion

### Blood parasite infection in house sparrows

Our study, conducted in urban and rural wild populations of house sparrows, provides several pieces of information regarding blood parasite infection in this species. The number of

infected individuals was not correlated with the urbanisation score defined in our four populations. Similarly, other previous studies did not find differences in prevalence between urban and rural habitats [38], even in the same biological model [7]. However, other contrasting results prevent us from suggesting a conclusive pattern, especially because previous studies found that avian malaria prevalence can differ between habitats and can be related to the degree of urbanisation [35–37, 39–42]. Changes in climatic conditions (mainly temperature and precipitation) and habitat characteristics can affect both vector-borne parasites and their insect vectors (e.g. reproductive cycle), and thus affect prevalence [36, 55, 75–77]. However, the populations investigated in this study are probably too close geographically to represent marked differences in climate (but see for instance [78]). One might also argue that the sampled cities are not large enough to observe any effect of urbanisation on parasite infection. Yet, previous studies reported morphological and physiological differences between the same urban and rural populations [44, 49]. Accordingly, at a larger scale of sampling, the degree of urbanisation did not correlate with *Plasmodium* prevalence in house sparrows either, even if larger cities were included [7]. Altogether, these studies seem to indicate that malaria prevalence would not be strongly affected by urbanisation in house sparrows, in our study system.

If prevalence can be affected by climatic conditions, it is expected that seasonality would explain a significant amount of the variation in malaria prevalence, especially in temperate areas, where vector abundance and susceptible host availability are likely to vary [79, 80]. Indeed, we observed that the number of infected individuals increased during the breeding season (May-August). This could be explained by an increase in vector abundance in the summer, as well as an increase in the number of fledglings and juveniles that are susceptible to being primo-infected [81, 82]. This latter hypothesis seems to be confirmed by an overall increase in the number of infected individuals during the course of the breeding season and by a higher number of infected individuals in juveniles than in adults, which has also been observed in previous studies [7, 83].

## Relationships between parasite infection and physiology

In a previous companion article using this data set, we demonstrated that physiology was not dramatically affected by the degree of urbanisation [44]. In this study, we examined avian malaria infection status and we specifically aimed to test its impact on physiology in house sparrows living in urban and rural places. We did not detect any physiological (i.e. haematocrit, baseline and stress-induced corticosterone) differences between infected and uninfected birds. Some studies, also in natural population, have investigated the association between physiological variables and blood parasite infection, and they produced mixed results: blood parasite infection was associated with detrimental effects on physiology in some but not all studies [20, 80, 84–89].

Several previous studies observed a lower haematocrit in infected birds compared to uninfected birds in multiple species, such as the red-winged blackbird [17, 20, 84, 85, 90]. Haematocrit is usually considered as a relevant marker integrating both red cell damage due to parasites, as well as costs of the immune response [90, 91], including destruction of like red blood cells by T-cell activity [92, 93]. In agreement, our analyses detected a trend for infected juveniles to have a lower haematocrit than uninfected juveniles, but the absence of significance prevents any conclusion. It is possible that heavily infected juveniles with a low haematocrit die quickly and thus were not sampled.

In previous studies, experimental elevation of corticosterone reduced parasite resistance (e.g. for avian malaria: [94, 95]), possibly due to the immunosuppressive effect of glucocorticoids [96]. However, this result has not always been confirmed in observational studies [80, 85,

86]. Other studies also found that higher corticosterone levels increased haemosporidian parasite tolerance [85], possibly due to tissue repair enhancement and/or damage limitations [96–98]. In addition, malaria could also lead to energetic costs, which may translate into increased corticosterone levels [99–101]. Here, we did not find any relationship between malaria infection and baseline or stress-induced corticosterone levels, suggesting that Hypothalamic-pituitary-adrenal axis function is not dramatically affected by malaria infection. In line with the present study, previous investigations often failed to detect any associations between infection and corticosterone levels [20, 85, 102], while significant, but mixed results were found when investigating the relationship between corticosterone levels and parasite intensity [80, 85, 86].

## Relationships between parasite infection, condition, and morphology

In our previous companion article [44], we demonstrated that the degree of urbanisation had an effect on some morphological variables and on some proxies of body condition. Here, we examined whether malaria infection status could affect these morphological variables in urban and rural house sparrows. According to the significant interactions that we detected, the relationships between blood parasite infection and morphology and condition appear to be complex. Intricate and inconsistent findings were previously reported in several studies [6, 7, 14, 17, 20, 84, 89, 90, 103, 104]. For example, Jiménez-Peñuela et al. found that infected house sparrows are in better condition than uninfected ones in wild populations [45], while Marzal et al. found the opposite pattern in wild house martins [27]. In addition, experimental captive studies did not find relationship between avian malaria infection status in several bird species [20, 105]. This may be because hosts in good condition may be able to allocate enough energy to trigger efficient immune responses [106, 107]; however, good condition may provide more resources for the parasite, which could increase virulence without leading to mortality [108–110]. Overall, our results indicate that infected juveniles are larger and in better condition than uninfected juveniles, and a similar result was found in another study on the same species [45]. Blood parasites may be associated with an increased risk of mortality [21, 23, 83], especially during primo infection, and this risk may increase when juveniles are in poor condition or not fully grown (small and light individuals, [16, 18, 111]). This may explain why we, unexpectedly, found that infected juveniles were larger and in better condition that uninfected ones. Conversely, in adults, infected birds were overall in poorer condition than uninfected individuals, suggesting that infection may have energetic costs (immune response, lower activity and feeding, and higher metabolism [17, 22, 95]).

## Underlying hypotheses

The absence of strong detectable effects of parasite infections on house sparrow physiology, morphology, and condition could be explained by three main hypotheses. First, the seasonality of blood parasite emergence could create co-variation with some factors, such as bird age, chick emancipation and sampling date, which could decrease statistical power and mask existing effects. Second, the sampled populations could be highly tolerant to the parasite, which prevent the detection of any effects, although the regional cause of such potential tolerance remains difficult to explain. Tolerance could be an adaptive strategy to minimize the negative impact of infection and the cost of resistance, and should be particularly suitable for endemic and chronic diseases [112, 113], such as haemosporidian parasites in the studied area [1, 114]. However, the balance between tolerance and resistance is still poorly understood for these parasites [85, 89, 95]. Third, strong selection pressures imposed by blood parasites could also explain the absence of results. In field studies, only birds that are able to fly can be caught and sampled. Infected birds that exhibit substantial disease symptoms may be completely absent

from the study due to considerably reduced their activity levels [15] or because they died quickly after infection [21]. As suggested by Jiménez-Peñuela et al. in their study [45], the remaining infected birds sampled could be the highest quality individuals, in which physiological costs of infection are below the detection threshold. In support with this hypothesis, one of the few significant results that we found was that infected juveniles were in better condition than uninfected juveniles, such as in [45]. This makes sense because juveniles are more likely to be primo-infected, and infected at higher intensities, than adults [1]. Furthermore, studies conducted in the UK observed that house sparrow declines are associated with reduced juvenile recruitment linked to lower overwinter survival [115, 116] and that the intensity of blood parasite infection reduced survival, even if this effect was also detected in adults [34]. Therefore, it would also be interesting to measure parasite intensities in our house sparrow samples to test how the parameters measured are correlated with parasitaemia and would corroborate the study conducted in the UK [34].

## Conclusion

To conclude, this study highlights the complexity of avian haemosporidan parasite dynamic as well as the difficulty to detect potential associated costs, especially in natural populations. Seasonality, selective disappearance of infected individuals of different quality, age or sex, and habitat-associated factors could interact and reduce our capacity to detect the effects of blood parasites, if there are any. Thus, the statistical power, limited in our study, has to be large enough to be able to test (and detect) these multiple interactions. Moreover, avian haemosporidian parasites consist of three genera comprising many species and strains [68, 117, 118], which could be different in term of biology, ecology and virulence [1, 119]. Co-infections between two or more blood parasite lineages could also have amplified negative effects [120, 121]. Considering these different levels of diversity would improve our understanding about these parasites and their effects on wild populations. In natural populations, longitudinal sampling during the course of the year (e.g. including the non-breeding season [84]) could provide valuable information on parasite dynamics and associated selective processes. A longitudinal survey could also be carried out using experimental approaches that simulate different environments (e.g. urban versus rural) and could provide knowledge about the evolutionary ecology of this very common, but still poorly understood, wild bird parasite.

## Supporting information

**S1 Fig. (A) Body mass and (B) tarsus length in relation to infection status (0 = uninfected, 1 = infected) and sex in adult house sparrows.**
(DOCX)

**S2 Fig. Body condition in relation to population and infection status in adult house sparrows.**
(DOCX)

**S1 Data.**
(CSV)

## Acknowledgments

We are very grateful to the "Mairie de Niort" for their support and for allowing us to capture birds in the city. We warmly thank Dr G. Sorci for providing us positive house sparrow samples for *Plasmodium*. We thank for her help with DNA extraction and Dr S. Hope for carefully

editing this manuscript. We thank G. Gouchet, L. Sourisseau and D. Dion for their help in the field; as well as S. Ruault, C. Trouvé and E. Beaugeard for their help with DNA extraction, molecular sexing and hormone assays.

## Author Contributions

**Conceptualization:** Coraline Bichet, François Brischoux, Frédéric Angelier.

**Data curation:** Coraline Bichet, François Brischoux, Alizée Meillère.

**Formal analysis:** Coraline Bichet.

**Funding acquisition:** Frédéric Angelier.

**Investigation:** Coraline Bichet, Frédéric Angelier.

**Methodology:** Coraline Bichet, Cécile Ribout, Charline Parenteau.

**Validation:** Coraline Bichet, Frédéric Angelier.

**Writing – original draft:** Coraline Bichet, Frédéric Angelier.

**Writing – review & editing:** Coraline Bichet, François Brischoux, Cécile Ribout, Charline Parenteau, Alizée Meillère, Frédéric Angelier.

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
