## [Decision Letter · Decision Letter 0]

12 Mar 2020

PONE-D-20-04379

Physiological and morphological correlates of blood parasite prevalence in urban and non-urban house sparrow populations

PLOS ONE

Dear Dr. Bichet,

Thank you for submitting your manuscript to PLOS ONE. After careful consideration, we feel that it has merit but does not fully meet PLOS ONE’s publication criteria as it currently stands. Therefore, we invite you to submit a revised version of the manuscript that addresses the points raised during the review process.

The review process is now complete, and three thorough reviews from highly qualified referees are included at the bottom of this letter. All reviewers including myself agree the manuscript deserves to be published. Although there is considerable merit in your paper, we also identified some concerns that must be considered in your resubmission. I particularly agree with the reviewers that the major limitation of the study is the small sampling size, where only 113 individual were analyzed, reducing the statistical power. Please, consider to discuss this study limitation accordingly. Another point must be better discussed is the fact that authors analyzed the overall prevalence of haemosporidian infection, not taking account the differences between Plasmodium and Haemoproteus genera regarding their biological parameters, specificity and virulence.

We would appreciate receiving your revised manuscript by Apr 26 2020 11:59PM. To enhance the reproducibility of your results, we recommend that if applicable you deposit your laboratory protocols in protocols.io, where a protocol can be assigned its own identifier (DOI) such that it can be cited independently in the future. For instructions see: http://journals.plos.org/plosone/s/submission-guidelines#loc-laboratory-protocols

We look forward to receiving your revised manuscript.

Kind regards,

Érika Martins Braga, Ph.D.

Academic Editor

PLOS ONE

Journal Requirements:

2. We note that Figure 1 in your submission contain map images which may be copyrighted. All PLOS content is published under the Creative Commons Attribution License (CC BY 4.0), which means that the manuscript, images, and Supporting Information files will be freely available online, and any third party is permitted to access, download, copy, distribute, and use these materials in any way, even commercially, with proper attribution. For these reasons, we cannot publish previously copyrighted maps or satellite images created using proprietary data, such as Google software (Google Maps, Street View, and Earth). For more information, see our copyright guidelines: http://journals.plos.org/plosone/s/licenses-and-copyright.

Reviewers' comments:

Reviewer's Responses to Questions

**Comments to the Author**

1. Is the manuscript technically sound, and do the data support the conclusions?

Reviewer #1: Yes

Reviewer #2: Partly

Reviewer #3: Partly

2. Has the statistical analysis been performed appropriately and rigorously? 

Reviewer #1: No

Reviewer #2: Yes

Reviewer #3: I Don't Know

3. Have the authors made all data underlying the findings in their manuscript fully available?

Reviewer #1: Yes

Reviewer #2: Yes

Reviewer #3: Yes

4. Is the manuscript presented in an intelligible fashion and written in standard English?

Reviewer #1: Yes

Reviewer #2: Yes

Reviewer #3: No

5. Review Comments to the Author

Reviewer #1: In the study entitled "Physiological and morphological correlates of blood parasite prevalence in urban and non-urban house sparrow populations), authors analyse several bird condition variables and related them with the infection status by avian malaria parasites. The major limitation of this study is the small sampling size, where only 113 individual were capture, especially in the light of age categories: (68 adults and 45 juveniles). For this reason, authors work with total prevalence/infection status, joining data from Plasmodium and Haemoproteus infections. This fact could be biasing results from the study because different parasite have different environmental requirements and physiological variable can affect bird differentially. Maybe results from this study can vary if both parasite were studied individually. I can understand that, due to the small sample size, it was not possible, but maybe from other studied authors can increase their dataset due to, this are old data and authors also refer to previous studies.

I have some important concerns that should be included in the manuscript:

Title is referring to parasite prevalence, where the bid results are regarding infection status. Please, be aware of this and adapt the title.

I was wondering why the corresponding author is different from first author. There is any particular circumstance?

Line 20: “However, but we observed…” two negatives are contradictory.

Line 22: “on physiology, condition, or morphology” condition alone is inconsistent. Maybe authors would say “physiological condition”? Health condition? Bird body condition?

INTRODUCTION

Lines:78-80: “breeding season because vector populations (mosquitoes) should become larger and more active as ambient temperature increases”, please be aware and comment in the main text that different mosquito population differ in their seasonality and environmental/weather requirements. Also, not only ambient temperature is a driver of vector population, due to rainfall are fairly a primary factor in mosquito life cycle. You can discuss the period of breeding season in the country where study is performed in order to bring light to this issue.

Line 86: Please, define “health”? Status? Body condition?

MATERIAL AND METHODS

Line 112: separate “captured” from “113”

Line 125: authors are measuring the bird “weight”? They talk about body mass and sometimes it can lead to errors. Please, clarify that bird weight s referred as body mass thereafter.

Lines 129-130: “We determined the age of the birds (adult or juvenile) based on plumage characteristics”, all juveniles were determined only visually? Any molecular techniques were employed in those cases where no plumage characteristic were not present? May is quite early in the breeding season and not all juveniles may have develop plumage dimorphism. Also, in line 148 authors talk about “molecular sexing” and subsequently they specify the procedure. Please, be consistent and modify the sentence in line 129 adding “when possible” or “in those cases were it was not possible molecular techniques were employed “see below”, or similar.

Line 154: double space between “modified” and “the”.

Line 164: Statistical analyses: No recapture data were present? Authors could think about control for the bird ID (or the ring number) in their models by using GLMMs or LMMs (function glmer and lmer in lme4 package in r software) including the ID as random factor. Also, try to control for the sampling size (N) as another covariate in your models. This could be a better option in order to control for the limitation of the small sampling size.

First, GLM were used to explore factors affecting infection status, pay attention to words. In line 165 authors talk about “parasite prevalence” where they really are not working with prevalence I understand (if not, a LM should be fitted). If I am wrong and they have used parasite prevalence, please specify better in Methods, explaining where prevalence and where infection status were used.

Line 181: why authors considered first- and second-order interaction. Please explain/justify this decision in the text to help lector in following the statistical procedure.

Overall, how much was the explained variance (R2) of your models? That’s an important variable to take into account in order to deeply understand the power of the statistical model (and your conclusions). You can check function “r.squaredGLMM” or similar in R.

RESULTS

Line 198: I am still not sure if authors are working with prevalence, or infection status. This is a problem of all this first section of the Result. Please, clarify if prevalence was measured (in Methods) and used (in Results) in this first section, because later, in the second and third sections (line 213 and 222), “infection status” is used. Please, clarify it along the whole main text.

Also, overall prevalence/infection are used (Plasmodium + Haemoproteus)? Maybe results from this study can vary if both parasite were studied individually. I can understand that, due to the small sample size, it was not possible, but maybe joining data from other studies from the same study area can increase the database.

Line 212. Related to the results from this paragraph “Relationship between parasite infection and physiology”. Have you thought about problem regarding the small sampling site used? I think authors should control for the sampling site N in each population, in order to discern if your no-significant results depend on the few birds used in this study more than to a lack of association among variables investigated. Please, try to repeat the LMs adding the variable N as independent variable.

DISCUSSION

Line 266: “urban and rural, natural populations of house sparrows” authors introduce a new kind of habitat that previously was not mentioned, natural population, or they want to refer to wild bird? I think is the second option because along all the manuscript this is the first time they talk about the natural environment. Please, be careful and check it. Also, again, they misunderstanding the concept of parasite prevalence (line 268). Please, use “infection status”, this is more appropriate.

Line 273: “and their insect vectors” a noun is missing. Insect vectors biology? Insect vector reproduction cycle? Insect vector environmental requirements? …

Lines 281-282: “Altogether, these studies suggest that malaria prevalence is not strongly affected by urbanization in house sparrows”. Please, soften this sentence.

Also, need to know the amount of explained variance (R2) of the statistical models in order to understand the real importance of this study.

Line 297: references suggested about the mixed results reported about association between physiological variables and blood parasite infection are quite a lot. “[20,82,86–96]”. If authors want to show the wide difference among 13 studies they should discuss more deeply the conclusion of these studies in relation with results they have found.

Line 312: HPA abbreviation has not been described before.

Line 320: ending a sentence with “complex” and staring a new one with the same word is not recommended.

Line 321: again, too many references. Try to focus your findings in the light of a restricted number of studies.

Line 339-342: why your population could be highly tolerant to the parasite? There are some particular environmental characteristics in those area that can protect them or alter their physiological conditions?

REFERENCES

Line 383: double space between University and Lithuania.

Ref. number 3, 12, 13, 30, 72, and 99 doi information is missing.

Ref. 94 and 114, please check authors names and the use of capital letters.

Please check the use of Journal name abbreviations, it is not always respected (e.g. Ref. 102, 104, 123).

TABLES

Overall, in the GLMs and LMs result tables, Estimates should be shown in order to understand the sign of the relation. Also, in the table title, should be indicate if these are results from a GLM or a LM, and the sample size used in the model (N). Also, Figure 2 B is not so clear. Authors show the relation with the prevalence (it is prevalence or Infection status?) and in the 2B values are all around 0 and 1. Please, clarify and explain more this figure in the text and in the title.

Reviewer #2: In this manuscript authors highlights the complex study system of host-pathogens interactions in areas with different level of urbanization that compromise bird’s health status. Due to the population decline of these passerine and the increasement of altered environments, its highly important to carry on these kinds of studies. I consider it a very interesting study, even though, there is some questions that authors should take into account to do an optimal work:

Major comments.

1) As the authors point out at the beginning of the introduction, the avian haemosporidian parasites consist of three genera, but they only identify two of them. Their molecular analysis for parasite screening explained at material and methods, use primers that can’t identify the Leucocytozoon parasites and neither the coinfection between Plasmodium and Haemoproteus. Even though it is complicated to assess the Pla/Hae coinfection, not considering the Leucocytozoon infection could lead authors to underestimate the overall prevalence of avian malaria infections in these localities, as well as a possible coinfection between Leu and Pla/Hae.

2) Authors only consider the infection status as “infected or not infected”, but they didn’t take into account if the birds are infected by Plasmodium or Haemoproteus genus. In my opinion, this can introduce noise to their analysis as Plasmodium is a generalist parasite and Haemoproteus is more specific, the consequences of being infected by one genera or another could be different for the bird health status and affect the results. Also, as the authors point out, these parasite are transmitted by different vector, thus, the prevalence of each genera in the different environments can change affecting the results in prevalence dynamics and how birds health status is in the different environments due to their alteration, not only because climate conditions, as they point in lines 272-274 of the discussion.

3) Authors should clarify what they consider as “prevalence”, if it is referred to the prevalence of each population or other concept. How they introduce this variable in the analysis? Because in the most part of statistical analysis they were working with infection status. Thus, they are not considering prevalence dynamics but infection probability. I encourage them to review their work and clarify the concept that they use each time.

4) In lines 58-60, and lines 324-326, authors do not consider that there are other studies that found similar results as the ones they found. Please, check “Jiménez-Peñuela (2019) Science of the Total Environment 651: 3015-3022” where authors show that urban juveniles house sparrows infected by Plasmodium or Haemoproteus had higher body mass that the uninfected ones, and that could be due to a selective disappearance of the urban individuals with lower body condition that can’t face the synergic effects of urbanization and avian haemosporidian infection. Thus, it can explain and support the third hypothesis of the authors in the discussion.

5) Authors claim to have calculated the degree of urbanization (lines 118-121) of the study areas. However, no such variable was included in the statistical analysis, neither quantitively nor qualitatively, only showing the population ID name as categoric factor. They either include the level of urbanization where each individual was captured, that could influence some of the results. I highly recommend to introduce this new variable, or also the interaction between population and infection status to asses if living in urban and non-urban areas influence either the infection status, the prevalence or the physiological and morphological consequences of being infected and live in urbanized areas.

6) Due to the limitation of the sample size, and because of variables such as age or sex included in the models, statistical power could be affected and reduced. I highly recommend authors to include some statistical output such as the R2 of the final models, in order to show the statistical power of their analysis.

7) The title does not correspond with the mayor conclusion of the manuscript due to infection status, and not prevalence, is the variable analyzed. Please, rethink it.

Minor comments.

1) Authors should light some statements that they do during the discussion (as line 281-282).

2) The authors use a type II of Anova for Linear Models, but in those models, they include first- and second-order interactions. In this kind of models, it is more accurate to use type III Anova as this analysis contains significant interaction. Please, check the “help” in R with “?Anova” and explore this possibility because it could change their the results.

3) The keyword “blood parasites” is already present in the title. Try to take advantage from the keyworks in order to maximize the match success of your article in the web search. If you use the same word you have in the title you are missing a great opportunity for amplification. In addition, sort them alphabetically.

4) They write the word “naïve” at line 40 and “naïve” at line 4. Please, check the correct form.

5) Authors should include data about the repeatability of measuring tarsus and wing length as they indicate that the data were collected by different people and it has been shown in other studies to be affecting results.

6) How was calculated the model predictions used in the figure 2B? It should be clarified.

7) In line 125, should be “Then” instead of “The”. In line 153 is “genera” not “genus”. In line 273 is “vector-borne” and not “vector-born”.

8) Authors should homogenize the bibliography format.

9) I highly recommend you do not use different color for indicating the same things. For example, in figures 2 and 3 the colors indicating juveniles and adults. Also, avoid the same color for indicating different things, for example infected vs not infected and adults vs juveniles in the supplementary material figures.

Reviewer #3: This manuscript addresses potential effects of avian haemosporidians in house sparrows in areas with different disturbance levels. The general result is that there was no major effects due to parasitism in the studied populations and the authors did a nice review providing possible reasons for that.

One major concern is that for some results authors treat “marginally” significant differences (no statistical difference) as actual differences and in some parts of the results the take-home message in this sense is very hidden. For example, the text in lines 229-239 is a bit confusing. I understand that there is no difference in body weight and tarsus length between infected and uninfected adult sparrows, but this conclusion is buried in the paragraph. The main idea here and in other parts of the manuscript should be made clear.

On this note, in lines 298-303, if authors chose to use a P-value of 0.05 a priori, they can’t state that differences in hematocrit between infected and uninfected juvenile sparrows wasn’t due to chance. In other words, marginally significant is not significant, so authors can’t state that this particular finding agree with those many references cited. Alternatively, authors can discuss that there is some evidence that avian malaria is associated to reduced hematocrit levels then providing possible reasons for these contrasting results.

Another important point is that the reference 34 didn’t find that Plasmodium prevalence affect house sparrow survival, but they did find that parasitemia had an effect on that. Authors should be careful when using this reference to back up their hypotheses of avian malaria affecting house sparrows in the present study since they didn’t check parasitemia.

The manuscript should be revised for sentence structure and there are some punctuations errors and typos throughout. Maybe even more important, the manuscript is too wordy at some points and it definitely could be more concise. Also, including too many references does not make arguments stronger or more appealing, so maybe the number of references can be reduced as well?

Bellow I added some other points that could be addressed in the review of this manuscript.

L20 - Remove “but”

L21 – This “especially in juveniles” is a bit odd here. Authors need to be more precise stating that this effect was only detected in juveniles.

L33-37: I am not sure if avian haemosporidians can infect “any” bird species. Using a broad term here would be safer. I think that both sentences could be combined since some ideas are repeated and because the introduction is a bit too wordy.

L37-41: The comparison between both sentences is inaccurate. Avian malaria in Hawaii can’t be used to contrast the previous sentence (“Due to a long co-evolution with their hosts, it was initially suggested that these parasites would not have large negative effects on wild birds”).

L44: Parasitaemia itself does not cause deleterious effects. Deleterious effects are related to the destruction of red blood cells and to tissue damage during exo-erythrocytic development. These effects can be exacerbated in hosts with high parasitemia.

L66: Please clarify what kind of disturbance is this.

L84-86: Authors reported morphological and physiological differences between sparrows captured in urban and rural areas. What were these differences?

L75: Use “breeding season” instead of “breeding period” throughout the manuscript for consistency.

L123: What does “among others” mean? Please clarify.

L136-141: Please clarify whether the same procedures were applied for the samples obtained 3 min and 30 min after bird capture.

Why is it important to measure wing length since body condition was calculated based on tarsus length? Does it make sense to add wing length in the LMs? It would be better to remove this measurement from the models if there is no clear justification for keeping it.

Authors should remove the information on the increase in prevalence during catching period in lines 199-202 and keep this information only in the two paragraphs that follow in the text.

L212: I think that most subheadings should be “Relationships” since authors tested many variables in most sections.

Most of the result section is unnecessarily long. For instance, instead of stating that there was a significant difference (and if there was a difference, we don’t need to use the word “significant”), authors could just write what the differences were. Also, I don’t think that all means and statistics metrics need to be included in the text as long as the main information is in Tables 1 and 2.

L266: I would argue that the present study doesn’t provide information on blood parasite dynamics because the authors didn’t follow the infection in the same individual across different sampling points.

L271: “Contrasting” would be a better word here instead of “contradicting”.

L312: Please write HPA in full here.

6. PLOS authors have the option to publish the peer review history of their article (what does this mean?). If published, this will include your full peer review and any attached files.

Reviewer #1: No

Reviewer #2: No

Reviewer #3: Yes: Francisco Ferreira-Junior

---

## [Author Response · Author response to Decision Letter 0]

21 Apr 2020

PONE-D-20-04379

“Physiological and morphological correlates of blood parasite prevalence in urban and non-urban house sparrow populations”

Now entitled 

“Physiological and morphological correlates of blood parasite infection in urban and non-urban house sparrow populations”

- Our responses to comments are provided in bold and italic font. We also added a version of our manuscript where changes from the previous version appear highlighted in blue -

Dear Dr. Bichet,

Thank you for submitting your manuscript to PLOS ONE. After careful consideration, we feel that it has merit but does not fully meet PLOS ONE’s publication criteria as it currently stands. Therefore, we invite you to submit a revised version of the manuscript that addresses the points raised during the review process.

The review process is now complete, and three thorough reviews from highly qualified referees are included at the bottom of this letter. All reviewers including myself agree the manuscript deserves to be published. Although there is considerable merit in your paper, we also identified some concerns that must be considered in your resubmission. I particularly agree with the reviewers that the major limitation of the study is the small sampling size, where only 113 individual were analyzed, reducing the statistical power. Please, consider to discuss this study limitation accordingly. Another point must be better discussed is the fact that authors analyzed the overall prevalence of haemosporidian infection, not taking account the differences between Plasmodium and Haemoproteus genera regarding their biological parameters, specificity and virulence.

> We thank the academic editor who appreciated our study and think it deserves to be published. We now provided some metrics regarding the statistical power of our models (Tables 2, 3, S1 and S2) and we discuss this limitation (lines 366-367). We also discuss that we were only able to consider Plasmodium and Haemoproteus together (see the specific replies to the reviewers 1 and 2 and lines 367-372). We replied point by point to all the reviewers’ comments and we hope that our manuscript will provide entire satisfaction. 

We would appreciate receiving your revised manuscript by Apr 26 2020 11:59PM. To enhance the reproducibility of your results, we recommend that if applicable you deposit your laboratory protocols in protocols.io, where a protocol can be assigned its own identifier (DOI) such that it can be cited independently in the future. For instructions see: http://journals.plos.org/plosone/s/submission-guidelines#loc-laboratory-protocols

• A rebuttal letter that responds to each point raised by the academic editor and reviewer(s). This letter should be uploaded as separate file and labeled 'Response to Reviewers'.

• A marked-up copy of your manuscript that highlights changes made to the original version. This file should be uploaded as separate file and labeled 'Revised Manuscript with Track Changes'.

• An unmarked version of your revised paper without tracked changes. This file should be uploaded as separate file and labeled 'Manuscript'.

We look forward to receiving your revised manuscript.

Kind regards,

Érika Martins Braga, Ph.D.

Academic Editor

PLOS ONE

Journal Requirements:

2. We note that Figure 1 in your submission contain map images which may be copyrighted. All PLOS content is published under the Creative Commons Attribution License (CC BY 4.0), which means that the manuscript, images, and Supporting Information files will be freely available online, and any third party is permitted to access, download, copy, distribute, and use these materials in any way, even commercially, with proper attribution. For these reasons, we cannot publish previously copyrighted maps or satellite images created using proprietary data, such as Google software (Google Maps, Street View, and Earth). For more information, see our copyright guidelines: http://journals.plos.org/plosone/s/licenses-and-copyright.

> We remake the Figure 1 in order to solve the copyright issues.

Reviewers' comments:

Reviewer #1

In the study entitled "Physiological and morphological correlates of blood parasite prevalence in urban and non-urban house sparrow populations), authors analyse several bird condition variables and related them with the infection status by avian malaria parasites. The major limitation of this study is the small sampling size, where only 113 individual were capture, especially in the light of age categories: (68 adults and 45 juveniles). For this reason, authors work with total prevalence/infection status, joining data from Plasmodium and Haemoproteus infections. This fact could be biasing results from the study because different parasite have different environmental requirements and physiological variable can affect bird differentially. Maybe results from this study can vary if both parasite were studied individually. I can understand that, due to the small sample size, it was not possible, but maybe from other studied authors can increase their dataset due to, this are old data and authors also refer to previous studies.

> The reason we analysed Plasmodium and Haemoproteus together is not due to the small sample size, but due to the detection method. The nested PCR we used did not allow separating both genera. Due to grant issues, we were not able to conduct any sequencing in order to identify the parasite lineage. Unfortunately too, we are not able to add more data in this study and the previous studies we referred actually used the same data set. Plasmodium and Haemoproteus are two genera included many strains, which can also be different in their virulence and infection patterns, and co-infections are likely to occur but still not easy to detect. For instance, see the studies of Palinauskas et al 2011 (Experimental Parasitology) and 2018 (International Journal of Parasitology). So, even if we were able to detect Plasmodium and Haemoproteus separately, some problems could remain: what do to with co-infected individuals? How taking into account that co-infection is often under-estimated, and how to be sure that individuals are infected by the same strain? We now discussed this potential limitation (lines 367-372) and we had some details in the Material and Methods (lines 149-151, 153-157 and 161-162). 

I have some important concerns that should be included in the manuscript:

Title is referring to parasite prevalence, where the bid results are regarding infection status. Please, be aware of this and adapt the title.

> We apologize for this confusion and we now use “infection status” everywhere. The title is now “Physiological and morphological correlates of blood parasite infection in urban and non-urban house sparrow populations”.

I was wondering why the corresponding author is different from first author. There is any particular circumstance?

> The last author is conducting his research on the consequences of urbanization in house sparrows since many years now. Therefore, he appears more appropriate to be the corresponding author. All co-authors agreed with that. 

Line 20: “However, but we observed…” two negatives are contradictory.

> Corrected (line 20).

Line 22: “on physiology, condition, or morphology” condition alone is inconsistent. Maybe authors would say “physiological condition”? Health condition? Bird body condition?

> We modified by “physiological, morphological and condition indexes” (lines 22-23) because condition not only referred to the body condition index (fat and muscle scores were also measured). We also modified sentences in our Introduction in order to avoid confusion regarding all the parameters we measured (lines 79-82). 

INTRODUCTION

Lines 78-80: “breeding season because vector populations (mosquitoes) should become larger and more active as ambient temperature increases”, please be aware and comment in the main text that different mosquito population differ in their seasonality and environmental/weather requirements. Also, not only ambient temperature is a driver of vector population, due to rainfall are fairly a primary factor in mosquito life cycle. You can discuss the period of breeding season in the country where study is performed in order to bring light to this issue.

> We modified our sentence to fit with reviewer’s recommendations (lines 76-77). We also already mentioned the rainfall in our Discussion (lines 268-271) and we discussed further the role of seasonality (lines 280-289). 

Line 86: Please, define “health”? Status? Body condition?

> We replaced “Here, we aimed to test whether malaria infection status could (2a) affect the health of house sparrows” by “We tested whether infection status could (2a) affect the physiology, the morphology and the condition of house sparrows” (lines 81-82). 

MATERIAL AND METHODS

Line 112: separate “captured” from “113”

> Done (line 109).

Line 125: authors are measuring the bird “weight”? They talk about body mass and sometimes it can lead to errors. Please, clarify that bird weight s referred as body mass thereafter.

> Modified accordingly (lines 120-121). 

Lines 129-130: “We determined the age of the birds (adult or juvenile) based on plumage characteristics”, all juveniles were determined only visually? Any molecular techniques were employed in those cases where no plumage characteristic were not present? May is quite early in the breeding season and not all juveniles may have develop plumage dimorphism. Also, in line 148 authors talk about “molecular sexing” and subsequently they specify the procedure. Please, be consistent and modify the sentence in line 129 adding “when possible” or “in those cases were it was not possible molecular techniques were employed “see below”, or similar.

> It was always possible to differentiate between adults and juveniles, based on plumage characteristics. This is actually easy. Only the sex was determined by molecular techniques and only in juveniles because sexual dimorphism is only present in adult house sparrows. Therefore, we cannot make the corrections suggested by the referee because the age was always determined visually and never molecularly. We added the sample size of visual sexing (for adults, n = 68) and of molecular sexing (for juveniles, n = 45) lines 146-147. 

Line 154: double space between “modified” and “the”.

> Done (line 151).

Line 164: Statistical analyses: No recapture data were present? Authors could think about control for the bird ID (or the ring number) in their models by using GLMMs or LMMs (function glmer and lmer in lme4 package in r software) including the ID as random factor. Also, try to control for the sampling size (N) as another covariate in your models. This could be a better option in order to control for the limitation of the small sampling size.

> When an individual was recaptured, we immediately released it to avoid any useless additional stress for the bird. We now added this information on lines 125-126. Our dataset did not include any recapture, and therefore, no need to include the bird ID as a random effect. Regarding the second part of the comment, we deeply apologize but we did not understand what the referee means here. We did not see how adding the sample size of each population would help to control for our small sample size. In other words, we did not see why the sample size would influence the infection status of an individual. If the sample size in each population was very different (unbalanced sampling), we could add a weight for each individual depending in which population it was caught (weights option in glm formula), but this is not the case (31, 28, 24 and 30 individuals, Table 1). However, to reassure the reviewer, we tried these “weighted” models and they did not differ from our previous results (see the Tables below). We also wanted to mention here that we now provided the statistical power (R2) varying between 0.65 and 0.03) of our models, in the Tables 2, 3, S1 and S2, and we discussed this point (lines 366-367). 

Table 1. Relationship between sex, population and capture date, and blood parasite infection (n = 113).

Dependent variable Infection status

 Adults

Parameter LR Chi2 Df p-value

Sex 2.34 1 0.125

Age 6.13 1 0.013

Population 0.33 3 0.955

Capture date 5.58 1 0.018

Table 2. Relationship between blood parasite infection status, age, sex, and population, and (A) physiological parameters, (B) morphological measurements and (C) condition indices (n = 113).

A. 

Dependent variable Haematocrit Baseline corticosterone Stress-induced corticosterone

Parameter Sum of Square Df F value p-value Sum of Square Df F value p-value Sum of Square Df F value p-value

Infection status 6.27.10-4 1 0.32 0.570 0.48 1 0.07 0.791 69.2 1 1.36 0.247

Age 0.01 1 5.42 0.022 52.86 1 7.80 0.006 750.00 1 14.71 <0.001

Sex 1.40.10-5 1 0.01 0.931 12.30 1 1.82 0.181 246.00 1 4.83 0.030

Population 8.11.10-3 3 1.40 0.247 5.99 3 0.29 0.829 1877.40 3 12.28 <0.001

Capture date 0.02 1 11.88 0.001 51.85 1 7.66 0.007 2010.30 1 39.43 <0.001

Population:capture date - - - - 72.02 3 3.54 0.017 1294.70 3 8.47 <0.001

Age:capture date - - - - - - - - 263.50 1 5.17 0.025

Residuals 0.20 105 690.82 102 5148.80 101 

B. 

Dependent variable Body mass Tarsus length Wing length

Parameter Sum of Square Df F value p-value Sum of Square Df F value p-value Sum of Square Df F value p-value

Infection status 1.19 1 0.50 0.480 0.35 1 0.96 0.329 7.18 1 1.92 0.168

Age 65.07 1 27.52 <0.001 3.36 1 9.30 0.003 351.56 1 94.27 <0.001

Sex 0.37 1 0.16 0.695 0.20 1 0.57 0.455 111.84 1 29.99 <0.001

Population 71.28 3 10.05 <0.001 10.16 3 9.38 <0.001 3.16 3 0.28 0.838

Capture date 3.50 1 1.48 0.227 2.96 1 8.19 0.005 6.99 1 1.87 0.174

Age:capture date 25.65 1 10.85 0.001 - - - - - - - -

Infection status:age 0.06 1 0.03 0.872 1.57 1 4.35 0.040 - - - -

Infection status:sex 0.85 1 0.36 0.551 0.68 1 1.88 0.173 - - - -

Sex:age 0.23 1 0.10 0.756 0.25 1 0.68 0.410 - - - -

Population:capture date - - - - 3.55 3 3.28 0.024 - - - -

Infection status:age:sex 11.24 1 4.75 0.032 1.71 1 4.73 0.032 - - - -

Residuals 236.47 100 35.36 98 391.55 105 

C. 

Dependent variable Fat score Muscle score Body condition

Parameter LR Chi2 Df p-value LR Chi2 Df p-value Sum of Square Df F value p-value

Infection status 0.19 1 0.667 0.23 1 0.628 0.07 1 0.24 0.624

Age 3.45 1 0.063 0.37 1 0.542 0.42 1 1.41 0.237

Sex 0.02 1 0.879 0.33 1 0.568 0.08 1 0.26 0.611

Population 2.01 3 0.571 1.66 3 0.646 1.61 3 1.82 0.147

Capture date 0.81 1 0.367 1.04 1 0.307 1.48 1 5.03 0.027

Infection status:age - - - - - - 1.26 1 4.27 0.041

Residuals 30.59 104 

First, GLM were used to explore factors affecting infection status, pay attention to words. In line 165 authors talk about “parasite prevalence” where they really are not working with prevalence I understand (if not, a LM should be fitted). If I am wrong and they have used parasite prevalence, please specify better in Methods, explaining where prevalence and where infection status were used.

> We replaced “prevalence” by “infection” (line 165). The infection status (binary, 0 for uninfected, 1 for infected) was used in all analyses, never the prevalence. We apologize for this misunderstanding. 

Line 181: why authors considered first- and second-order interaction. Please explain/justify this decision in the text to help lector in following the statistical procedure.

> The required information is now provided lines 170-172, 180-183.

Overall, how much was the explained variance (R2) of your models? That’s an important variable to take into account in order to deeply understand the power of the statistical model (and your conclusions). You can check function “r.squaredGLMM” or similar in R.

> We now provided this information in the Tables 2, 3, S1, S2 (see also lines 197-198) and discussed this limitation (lines 366-367). The small (in general, but some are quite high: 0.65, 0.64, 0.49, 0,46, see tables) R2 we obtained in all our models is actually not very surprising since we hardly found any significant effects of our explanatory variables. The statistical power depends on the sample size, but also on the effect size of the factor investigated and on the random variation around this factor. These two last parameters are linked to biology and independent of the sample size.

RESULTS

Line 198: I am still not sure if authors are working with prevalence, or infection status. This is a problem of all this first section of the Result. Please, clarify if prevalence was measured (in Methods) and used (in Results) in this first section, because later, in the second and third sections (line 213 and 222), “infection status” is used. Please, clarify it along the whole main text.

> We always worked with the infection status of an individual, binary variable (0 or 1) in all our analyses. To avoid this misunderstood, we replaced prevalence by infection status when possible, otherwise we used the term “number of infected individuals” (e.g. lines 202, 205, 206, 208). 

Also, overall prevalence/infection are used (Plasmodium + Haemoproteus)? Maybe results from this study can vary if both parasite were studied individually. I can understand that, due to the small sample size, it was not possible, but maybe joining data from other studies from the same study area can increase the database.

> Please see our reply to a similar comment above.

Line 212. Related to the results from this paragraph “Relationship between parasite infection and physiology”. Have you thought about problem regarding the small sampling site used? I think authors should control for the sampling site N in each population, in order to discern if your no-significant results depend on the few birds used in this study more than to a lack of association among variables investigated. Please, try to repeat the LMs adding the variable N as independent variable.

> We believe that the reviewer is referring about “sample size” and not “sample site”. Please see our reply to a similar comment above.

DISCUSSION

Line 266: “urban and rural, natural populations of house sparrows” authors introduce a new kind of habitat that previously was not mentioned, natural population, or they want to refer to wild bird? I think is the second option because along all the manuscript this is the first time they talk about the natural environment. Please, be careful and check it. Also, again, they misunderstanding the concept of parasite prevalence (line 268). Please, use “infection status”, this is more appropriate.

> We replaced “natural” by “wild” (line 263). We also replace “We did not detect any differences in blood parasite prevalence among the four populations” by “The number of infected individuals did not differ among the four populations” (lines 264-265). 

Line 273: “and their insect vectors” a noun is missing. Insect vectors biology? Insect vector reproduction cycle? Insect vector environmental requirements? …

> Climatic conditions and habitat characteristics can affect the parasites, as well as the vectors (mosquitos in this case), in many aspects: abundance, richness, reproduction, survival… Since this is not the topic of our study, we do not want to provide details here, but we added in parentheses “e.g. reproductive cycle” (line 270). 

Lines 281-282: “Altogether, these studies suggest that malaria prevalence is not strongly affected by urbanization in house sparrows”. Please, soften this sentence.

> We modified “Altogether, these studies suggest that malaria prevalence is not strongly affected by urbanization in house sparrows” by “Altogether, these studies seem to indicate that malaria prevalence would not be strongly affected by urbanization in house sparrows” (lines 277-279). 

Also, need to know the amount of explained variance (R2) of the statistical models in order to understand the real importance of this study.

> Please see our reply to a similar comment above.

Line 297: references suggested about the mixed results reported about association between physiological variables and blood parasite infection are quite a lot. “[20,82,86–96]”. If authors want to show the wide difference among 13 studies they should discuss more deeply the conclusion of these studies in relation with results they have found.

> We modified the corresponding sentence accordingly (lines 293-294). We reduced the number of cited references at the end of the sentence (line 295). All these references are cited again in the following paragraphs and provide more information about their findings, in relation with our study (lines 296-302 for haematocrit and lines 303-315 for corticosterone). Overall, we reduced the number of cited studies (from 127 to 115), and we did not want to reduce it more since one of the reviewer appreciated the review we made about the different hypotheses which could explain our results (lines 334-359). 

Line 312: HPA abbreviation has not been described before.

> Corrected (line 311).

Line 320: ending a sentence with “complex” and staring a new one with the same word is not recommended.

> We replaced the second “complex” by “intricate” (line 319). 

Line 321: again, too many references. Try to focus your findings in the light of a restricted number of studies.

> We reduced the number of the cited studies here (line 321) but we believe it was important to keep a significant number since the main message here is that the results are inconsistent and we tried to give an explanation in the followed sentences (lines 321-332). 

Line 339-342: why your population could be highly tolerant to the parasite? There are some particular environmental characteristics in those area that can protect them or alter their physiological conditions?

> This is a potential hypothesis but we have no evidence for that yet, as for the other mentioned hypotheses (lines 334-359). As far as we know, these populations are not particular in term of environmental characteristics, but this is difficult to conclude since a lot of environmental parameters (food or partner availability, predators, pollution) and physiological parameters (immunity for example) still need to be investigated and compared between these populations (lines 340-341). It would also be difficult to find “reference” populations to compare with. Some populations can be more resistant to parasites and diseases depending on the health status of the individuals, the genetic background, the co-evolution time and others. On the other side, in particular cases, tolerance could be a strategy with reduced costs, in comparison to resistance. Mounting an immune response is costly in term of energy and produces by-products like oxidative stress and other immunopathological costs. In case of chronic disease, like avian malaria, it would be more adaptive to be tolerant instead of resistant, especially in populations where the parasite is always present at a significant prevalence (like our populations). We explained this strategy in lines 341-343, and it was difficult to provide more information here since the balance between tolerance and resistance is still poorly understood for these blood parasites (lines 344-345). 

REFERENCES

Line 383: double space between University and Lithuania.

> Corrected (line 392).

Ref. number 3, 12, 13, 30, 72, and 99 doi information is missing.

> Added (lines 396, 425, 429, 606, 655). We were not able to find any DOI associated with the reference 30.

Ref. 94 and 114, please check authors names and the use of capital letters.

> These references are now deleted in order to reduce the number of cited studies as suggested by the reviewers. 

Please check the use of Journal name abbreviations, it is not always respected (e.g. Ref. 102, 104, 123).

> Corrected (e.g. lines 662, 664, 706).

TABLES

Overall, in the GLMs and LMs result tables, Estimates should be shown in order to understand the sign of the relation. Also, in the table title, should be indicate if these are results from a GLM or a LM, and the sample size used in the model (N). 

> We followed the reviewer’s comment and modified our Tables (Table 3, S1 and S2) and their captions (lines 754-755, 756-758 and SI) accordingly. We also added a new table, Table 2 (the ancient Table 2 in now Table 3) providing the model parameters for the GLM testing the relationship between sex, population and capture date, and blood parasite infection.

Also, Figure 2 B is not so clear. Authors show the relation with the prevalence (it is prevalence or Infection status?) and in the 2B values are all around 0 and 1. Please, clarify and explain more this figure in the text and in the title.

> We modified the Figure 2B and the legend accordingly (lines 765-769). 

Reviewer #2:

In this manuscript authors highlights the complex study system of host-pathogens interactions in areas with different level of urbanization that compromise bird’s health status. Due to the population decline of these passerine and the increasement of altered environments, its highly important to carry on these kinds of studies. I consider it a very interesting study, even though, there is some questions that authors should take into account to do an optimal work:

> We thank the reviewer for these positive comments and we hope that this revised version will give entire satisfaction. 

Major comments.

1) As the authors point out at the beginning of the introduction, the avian haemosporidian parasites consist of three genera, but they only identify two of them. Their molecular analysis for parasite screening explained at material and methods, use primers that can’t identify the Leucocytozoon parasites and neither the coinfection between Plasmodium and Haemoproteus. Even though it is complicated to assess the Pla/Hae coinfection, not considering the Leucocytozoon infection could lead authors to underestimate the overall prevalence of avian malaria infections in these localities, as well as a possible coinfection between Leu and Pla/Hae.

> We actually tested all our individuals for Leucocytozoon and they were all negative. That is why we did not include them in the study. This is not surprising since infection by Leucocytozoon is rare in house sparrows (see for instance the database MalAvi: http://130.235.244.92/Malavi/ for a very nice overview of the lineage of blood parasite detected in many passerine hosts). We now indicated that we also tested for Leucocytozoon on lines 149-151 and 161-162. 

2) Authors only consider the infection status as “infected or not infected”, but they didn’t take into account if the birds are infected by Plasmodium or Haemoproteus genus. In my opinion, this can introduce noise to their analysis as Plasmodium is a generalist parasite and Haemoproteus is more specific, the consequences of being infected by one genera or another could be different for the bird health status and affect the results. Also, as the authors point out, these parasites are transmitted by different vector, thus, the prevalence of each genera in the different environments can change affecting the results in prevalence dynamics and how birds health status is in the different environments due to their alteration, not only because climate conditions, as they point in lines 272-274 of the discussion.

> It is true that the genera Plasmodium is more generalist but some strains of Haemoproteus are also able to infect a large number of species. Haemoproteus is actually the most commonly encountered (67% of infected bird species, while it is 42% for Plasmodium and 39 for Leucocytozoon). For example, the lineage TURDUS2 was detected in 22 host species from 16 avian genera, and the lineage PARUS1 in 24 species from 17 lineage (see the database MalAvi: http://130.235.244.92/Malavi/). Vectors from Culicoides genera mainly transmit Haemoproteus lineages, while Culicidae, Culex and Culiseta transmit mainly Plasmodium (Valkiunas, Iezhova 2004 Journal of Parasitology 90; Valkiūnas 2005 book from CRC press). We now mention that these parasites are transmitted by different vectors, but since vectors are not the topic of our study we did want to provide more details (line 33).

Unfortunately, our molecular method did not allow to differentiate between Plasmodium and Haemoproteus. We are also not able to conduct any sequencing to identify the strain. Even if, co-infections are very difficult to detect, and under-estimated in all studies conducted on wild populations and these parasites, even with sequencing. And even if we will be able to distinguish between Plasmodium and Haemoproteus, this is a quite non-ending story because different strains of Plasmodium can also have different virulence and infection dynamic and could differently affect the host. For instance, see the studies of Palinauskas et al 2011 (Experimental Parasitology) and 2018 (International Journal of Parasitology). We now discussed this potential limitation in the Discussion (lines 367-372). 

3) Authors should clarify what they consider as “prevalence”, if it is referred to the prevalence of each population or other concept. How they introduce this variable in the analysis? Because in the most part of statistical analysis they were working with infection status. Thus, they are not considering prevalence dynamics but infection probability. I encourage them to review their work and clarify the concept that they use each time.

> We apologize for this confusion and we now clarify this in the whole manuscript (for example in the title, lines 165, 202, 208, 264). We used the infection status (infected versus non-infected, binomial variable) in all our analysis. Only the Figure 2A presented the proportion of infected individual because we wanted to illustrate the differences in the proportion of infected individuals between populations (caption modified accordingly on lines 765-769). 

4) In lines 58-60, and lines 324-326, authors do not consider that there are other studies that found similar results as the ones they found. Please, check “Jiménez-Peñuela (2019) Science of the Total Environment 651: 3015-3022” where authors show that urban juveniles house sparrows infected by Plasmodium or Haemoproteus had higher body mass that the uninfected ones, and that could be due to a selective disappearance of the urban individuals with lower body condition that can’t face the synergic effects of urbanization and avian haemosporidian infection. Thus, it can explain and support the third hypothesis of the authors in the discussion.

> We thank the reviewer to provide the reference of this very interesting study. We now included it in our manuscript (lines 325-326 and line 62). 

5) Authors claim to have calculated the degree of urbanization (lines 118-121) of the study areas. However, no such variable was included in the statistical analysis, neither quantitively nor qualitatively, only showing the population ID name as categoric factor. They either include the level of urbanization where each individual was captured, that could influence some of the results. I highly recommend to introduce this new variable, or also the interaction between population and infection status to asses if living in urban and non-urban areas influence either the infection status, the prevalence or the physiological and morphological consequences of being infected and live in urbanized areas.

> The degrees of urbanisation were calculated in a previous study (Meilliere et al 2015 PlosOne 10(8)) using the same dataset and were 2.10 for La Rochelle, 1.61 for Niort, -1.21 for Villefollet and -2.50 for CEBC. With only four points presenting such dichotomy and no gradient, we preferred a qualitative approach over a quantitative approach. Introducing a qualitative variable (populations type: urban versus rural) in our analyses will create a nested factor (population ID is nested in population type) and we did not want to over-parametrized our models. Moreover, the influence of urbanization on physiological parameter is not the topic of our study and we added population ID to control for population differences in physiological and morphological parameters already found in the previous study (Meilliere et al 2015 PlosOne 10(8)). However, we tested if the proportion of infected individuals was different between rural and urban populations (line 203) and it was not the case. To avoid any further confusion, we deleted all information related with urbanisation gradient in our manuscript (line 115 and Table 1).

6) Due to the limitation of the sample size, and because of variables such as age or sex included in the models, statistical power could be affected and reduced. I highly recommend authors to include some statistical output such as the R2 of the final models, in order to show the statistical power of their analysis.

> We now provided this information on the Tables 2, 3, S1 and S2 (see also lines 197-198), and discussed the limitation links to the power of our analysis (lines 365-366). The small (in general, but some are quite high: 0.64, 0.49, 0,46, see tables) R2 we obtained in all our models is actually not very surprising since we hardly found any significant effects of our explanatory variables. The statistical power depends on the sample size, but also on the effect size of the factor investigated and on the random variation around this factor. These two last parameters are linked to biology and independent of the sample size.

7) The title does not correspond with the major conclusion of the manuscript due to infection status, and not prevalence, is the variable analyzed. Please, rethink it.

> The study is now entitled “Physiological and morphological correlates of blood parasite infection in urban and non-urban house sparrow populations”.

Minor comments.

1) Authors should light some statements that they do during the discussion (as line 281-282).

> Agree. We modified “Altogether, these studies suggest that malaria prevalence is not strongly affected by urbanization in house sparrows” by “Altogether, these studies seem to indicate that malaria prevalence would not be strongly affected by urbanization in house sparrows” (lines 277-279).

2) The authors use a type II of Anova for Linear Models, but in those models, they include first- and second-order interactions. In this kind of models, it is more accurate to use type III Anova as this analysis contains significant interaction. Please, check the “help” in R with “?Anova” and explore this possibility because it could change their the results.

> We now used type III Anova as now stated line 193. It did not modify any results (see Result section). 

3) The keyword “blood parasites” is already present in the title. Try to take advantage from the keyworks in order to maximize the match success of your article in the web search. If you use the same word you have in the title you are missing a great opportunity for amplification. In addition, sort them alphabetically.

> We replaced blood parasites by “avian malaria” in the keywords and ordered them by alphabetic order (lines 29-30).

4) They write the word “naïve” at line 40 and “naïve” at line 4. Please, check the correct form.

> We believe the reviewer wanted to refer to line 44 and not line 4. Corrected (line 38).

5) Authors should include data about the repeatability of measuring tarsus and wing length as they indicate that the data were collected by different people and it has been shown in other studies to be affecting results.

> Morphological and haematocrit measures were only made by A.M.. We added this information lines 121 and 129.

6) How was calculated the model predictions used in the figure 2B? It should be clarified.

> We apologize for this misunderstood. We now changed the name of the y-axis by “infection status” and modified the caption of this figure accordingly (lines 764-768). 

7) In line 125, should be “Then” instead of “The”. In line 153 is “genera” not “genus”. In line 273 is “vector-borne” and not “vector-born”.

> Corrected accordingly (lines 120, 154, 270).

8) Authors should homogenize the bibliography format.

> We corrected the mistakes and hope everything is in order now. 

9) I highly recommend you do not use different color for indicating the same things. For example, in figures 2 and 3 the colors indicating juveniles and adults. Also, avoid the same color for indicating different things, for example infected vs not infected and adults vs juveniles in the supplementary material figures.

> We modified the Figures accordingly. In the Figure 1, urban populations are now representing by brown dots, while rural populations by green dots. In the Figures 2 and 3, the black colour still refers to the adults, while the grey colour represents the juveniles. In the Figure S1, males are represented by the blue colour, while females by the red colour. In the Figure S2, infected individuals are represented by the orange colour, while uninfected individuals by the black colour. 

Reviewer #3: 

This manuscript addresses potential effects of avian haemosporidians in house sparrows in areas with different disturbance levels. The general result is that there was no major effects due to parasitism in the studied populations and the authors did a nice review providing possible reasons for that.

> We thank the reviewer for this positive comment. We are grateful that our review in the discussion was appreciated.

One major concern is that for some results authors treat “marginally” significant differences (no statistical difference) as actual differences and in some parts of the results the take-home message in this sense is very hidden. For example, the text in lines 229-239 is a bit confusing. I understand that there is no difference in body weight and tarsus length between infected and uninfected adult sparrows, but this conclusion is buried in the paragraph. The main idea here and in other parts of the manuscript should be made clear.

> We apologize for these confusions and we now clarified these points (e.g. lines 224, 229, 242, 249). 

On this note, in lines 298-303, if authors chose to use a P-value of 0.05 a priori, they can’t state that differences in hematocrit between infected and uninfected juvenile sparrows wasn’t due to chance. In other words, marginally significant is not significant, so authors can’t state that this particular finding agree with those many references cited. Alternatively, authors can discuss that there is some evidence that avian malaria is associated to reduced hematocrit levels then providing possible reasons for these contrasting results.

> We modified our discussion accordingly (lines 217-218 and 299-302). 

Another important point is that the reference 34 didn’t find that Plasmodium prevalence affect house sparrow survival, but they did find that parasitemia had an effect on that. Authors should be careful when using this reference to back up their hypotheses of avian malaria affecting house sparrows in the present study since they didn’t check parasitemia.

> We modified this point accordingly and we specified that this study found that parasitaemia affects house sparrow survival (lines 51, 357-359). 

The manuscript should be revised for sentence structure and there are some punctuations errors and typos throughout. Maybe even more important, the manuscript is too wordy at some points and it definitely could be more concise. 

> We apologize for this and we are actually quite surprized by this comment since the manuscript was carefully editing by a native English speaker (Dr. S. Hope, see our acknowledgement section). We corrected some typos (e.g. lines 120, 154, 270) and we revised sentence structure (e.g. lines 33, 46, 270, 283). We hope that the manuscript is now better written and will satisfy the reviewer. 

Also, including too many references does not make arguments stronger or more appealing, so maybe the number of references can be reduced as well?

> We reduced the number of references (from 127 to 115), but as the reviewer positively pointed out, our aim was also to make a review of the different hypotheses which could explain our results. Therefore, to make this review meaningful and exhaustive, it seems important to us to keep a significant number of cited papers.

Bellow I added some other points that could be addressed in the review of this manuscript.

L20 - Remove “but”

> Removed (line 20).

L21 – This “especially in juveniles” is a bit odd here. Authors need to be more precise stating that this effect was only detected in juveniles.

> Actually, we found that, with the model included all individuals (adults and juveniles) that the proportion of infected individuals increased during the catching period (line 204). When juveniles and adults were separated into two different models, we found this significant effect only in the model included the juveniles (SI Table S1). But to avoid any misunderstood, we modified the sentence “However, we observed that prevalence increased during the course of the season, especially in juveniles, which also exhibited lower prevalence than adults” by “However, we observed that the proportion of infected individuals increased during the course of the season, and that juveniles exhibited lower prevalence than adults” (lines 20-22).

L33-37: I am not sure if avian haemosporidians can infect “any” bird species. Using a broad term here would be safer. I think that both sentences could be combined since some ideas are repeated and because the introduction is a bit too wordy.

> Agree. We replaced “Avian haemosporidian parasites consist of three genera (Plasmodium, Haemoproteus, Leucocytozoon) that are transmitted by insect vectors to birds [1]. These blood parasites can infect many bird species and have been extensively studied in the last few decades as a model in evolutionary ecology [2]” by “Avian haemosporidian parasites consist of three genera (Plasmodium, Haemoproteus, Leucocytozoon) that are transmitted by different insect vectors to birds [1], and have been extensively studied in the last few decades as a model in evolutionary ecology [2]. Avian haemosporidian parasites can be found in numerous species worldwide and their prevalence can reach very high percentages in some wild bird populations [3–9]” (lines 32-36). 

L37-41: The comparison between both sentences is inaccurate. Avian malaria in Hawaii can’t be used to contrast the previous sentence (“Due to a long co-evolution with their hosts, it was initially suggested that these parasites would not have large negative effects on wild birds”).

> We modified the sentence “However, it has now been demonstrated that these parasites can have a huge detrimental effect on bird populations, especially following their introduction to naive populations” by “On the other hand, in naive populations, these parasites can have a huge detrimental effect on bird populations” (line 38).

L44: Parasitaemia itself does not cause deleterious effects. Deleterious effects are related to the destruction of red blood cells and to tissue damage during exo-erythrocytic development. These effects can be exacerbated in hosts with high parasitemia.

> We replaced “The deleterious effects of parasitaemia can be serious during this phase, especially for naive hosts” by “Following the primo infection, the host experiences an acute phase with a high level of parasitaemia [1], with important red blood cell destructions and tissue damages caused by the parasite developments” (lines 41-44).

L66: Please clarify what kind of disturbance is this.

> Clarified (line 65). 

L84-86: Authors reported morphological and physiological differences between sparrows captured in urban and rural areas. What were these differences?

> Details are now provided (line 84). 

L75: Use “breeding season” instead of “breeding period” throughout the manuscript for consistency.

> Modified (lines 72, 205, 210, 214, 250, 291).

L123: What does “among others” mean? Please clarify.

> We deleted “among others” (line 120), and we add the sentence “The blood collected will also be used to measure haematocrit and for molecular analyses (see below)” (lines 117-118).

L136-141: Please clarify whether the same procedures were applied for the samples obtained 3 min and 30 min after bird capture.

> Clarified (lines 134-135).

Why is it important to measure wing length since body condition was calculated based on tarsus length? Does it make sense to add wing length in the LMs? It would be better to remove this measurement from the models if there is no clear justification for keeping it.

> Wing length in another measure of morphology providing different information than tarsus length. Tarsus grows mainly during early life stages (chick and juvenile), and its length stays stable during adulthood. Wing length reflects feathers growth, which occurs every year, during the all life of the bird. As such, tarsus length would reflect development (and associated constraints) at chick/juvenile stage, while wing length would reflect feather growth (and associated constraints) at all life stages (juvenile and adult). Therefore, we believe we should keep both tarsus length and wing length in our analyses. 

Authors should remove the information on the increase in prevalence during catching period in lines 199-202 and keep this information only in the two paragraphs that follow in the text.

> We respectfully disagree. The first paragraph referred to the results obtained by the model including all the birds together. The second paragraph referred to the results obtained by the model including only the juveniles, while the third paragraph referred to the results obtained by the model including only the adults. So, we believe that all these results are important.

L212: I think that most subheadings should be “Relationships” since authors tested many variables in most sections.

> Corrected (lines 212, 222, 247, 291, 317).

Most of the result section is unnecessarily long. For instance, instead of stating that there was a significant difference (and if there was a difference, we don’t need to use the word “significant”), authors could just write what the differences were. 

> Modified accordingly (e.g. lines 224, 229, 242, 249, 253, 256).

Also, I don’t think that all means and statistics metrics need to be included in the text as long as the main information is in Tables 1 and 2.

> According to this comment and to the other reviewer’s comments, we modified our tables, which now provide the model summaries (with estimates). We simplified and shortened the text in the result section by avoiding to five the model parameters already presented in the Tables, and only mentioned means and metrics like ANOVA results when it was meaningful to mention (e.g. lines 202-203, 225-226, 230-231, 243). 

L266: I would argue that the present study doesn’t provide information on blood parasite dynamics because the authors didn’t follow the infection in the same individual across different sampling points.

> We modified accordingly (line 264). 

L271: “Contrasting” would be a better word here instead of “contradicting”.

> Modified (line 267). 

L312: Please write HPA in full here.

> Done (line 311).

---

## [Decision Letter · Decision Letter 1]

6 May 2020

PONE-D-20-04379R1

Physiological and morphological correlates of blood parasite infection in urban and non-urban house sparrow populations

PLOS ONE

Dear Dr. Bichet,

Thank you for submitting your manuscript to PLOS ONE. After careful consideration, we feel that it has merit but does not fully meet PLOS ONE’s publication criteria as it currently stands. Therefore, we invite you to submit a revised version of the manuscript that addresses the points raised during the review process.

The reviewers and I came to the consensus that the manuscript still deserves attention. Two reviewers point out that the authors did not respond to the questions raised adequately. I strongly agree with them and emphasize that the authors must dedicate to answer the points raised with the utmost precision and to make all those reconsiderations in the manuscript to be submitted.

We would appreciate receiving your revised manuscript by Jun 20 2020 11:59PM. To enhance the reproducibility of your results, we recommend that if applicable you deposit your laboratory protocols in protocols.io, where a protocol can be assigned its own identifier (DOI) such that it can be cited independently in the future. For instructions see: http://journals.plos.org/plosone/s/submission-guidelines#loc-laboratory-protocols

We look forward to receiving your revised manuscript.

Kind regards,

Érika Martins Braga, Ph.D.

Academic Editor

PLOS ONE

Reviewers' comments:

Reviewer's Responses to Questions

**Comments to the Author**

1. If the authors have adequately addressed your comments raised in a previous round of review and you feel that this manuscript is now acceptable for publication, you may indicate that here to bypass the “Comments to the Author” section, enter your conflict of interest statement in the “Confidential to Editor” section, and submit your "Accept" recommendation.

Reviewer #1: (No Response)

Reviewer #2: (No Response)

Reviewer #3: All comments have been addressed

2. Is the manuscript technically sound, and do the data support the conclusions?

Reviewer #1: Partly

Reviewer #2: Partly

Reviewer #3: Yes

3. Has the statistical analysis been performed appropriately and rigorously? 

Reviewer #1: No

Reviewer #2: No

Reviewer #3: Yes

4. Have the authors made all data underlying the findings in their manuscript fully available?

Reviewer #1: Yes

Reviewer #2: Yes

Reviewer #3: Yes

5. Is the manuscript presented in an intelligible fashion and written in standard English?

Reviewer #1: Yes

Reviewer #2: Yes

Reviewer #3: Yes

6. Review Comments to the Author

Reviewer #1: In this new version, authors have been improved the contents of the manuscript, addressing some of the queries proposed by reviews, although I think most of the ‘content’ changes are still missing. The authors have addressed the majority of the queries proposed with a too vague and not entirely conclusive tone. Also, I have noticed that authors avoided to give concrete answers to important and fundamental questions from a data analysis point of view.

The main problem I still see, is the lack of a complete model were all independent variables (physiological, morphological, and condition indices) are analysed at the same time by using infection status as dependent. Authors perform several little GLMs were they test different variables with infectious status but the overall effect is missing. Which are the most important variables? Which variable are driving infection in sparrows? I suggest this because the reader too often gets lost among so many non-significant results, tables, fragmented figures, supplementary information with significant results, sometimes some of them nor discussed in the manuscript.

I can suggest a general model with all the independent variables together in a AIC selection in order to have a more complete vision of the reality. In this model, authors should include a categorical variable of the sampling site (no size), as I previous commented them in the first review (but that authors said did not understand).

Results about no differences between habitat category (urban/rural), should be also taken with care. As I suggest previously, habitat category (urban/rural) or sampling site (the four localities, as categorical), should have been a variable took into account in the other GLMs, and used as random factor (glmer function in R) when authors want to get pattern about independent variables that could be driving infection in birds, disengaging from environmental patterns.

Also, the manuscript has still lots of minor weaknesses that were pointed out in the text I attached.

Reviewer #2: Authors correctly clarify the most part of the issues raised in the first major revision, but they still have considerable points that were not solved in the first time, or needs to be clarify even more (previous point 3 to 5). In detail:

3) Authors should have standardized the use about “infection status or probability” terms. They have corrected it in the most part of the manuscript, but in the first part of results (lines 201-210), they still refer to “The number of infected individuals” which is not correct as in the table cited they use “infection status”. Moreover, in the figure 2B, they labelled the Y axis as “infection status” that is supposed to be a binary variable, but the axis has intermediate value. Why? Please, correct this ‘again’ and make sure that you standardize it all along the manuscript, including figures, tables, and Supplementary Information.

4) Authors were asked to consider the results from Jiménez-Peñuela et al. 2019 in their study and even they include it in line 62 and 326, they did not consider it in their third hypothesis proposed in the discussion, that is the same hypothesis proposed in this work as I said in the first review. Moreover, in lines 66-68 they made an inappropriate statement “has never been examined in house sparrow”, but during all the introduction they are referring to works that already have evaluated similar variables in house sparrows, including in the suggested reference. The same occur with the statement on lines 277-278, where I suggested softening the tone in minor comments. Please, correct any similar statements along all the manuscript ‘again’ and make sure to check what other authors have recently did in the same topic, especially the ones that you are referring along your work.

5) Authors were asked to include either a qualitative or quantitative variable evaluating the urbanization of their population, due to as I said, including population ID in the statistical analysis only indicates if there was any difference between populations, but it did not compare urban and non-urban ones. Authors are claiming through all the manuscript that they compare urban and non-urban population (title, keywords, introduction, discussion…) but their answer to this request was: “The influence of urbanization on physiological parameter is not the topic of our study and we added population ID to control for population differences in physiological and morphological parameters already found in the previous study (Meilliere et al 2015 PlosOne 10(8))”.

I don't entirely agree with that answer and I can only comprehend two options: 1) if you do not include any variable that evaluate urbanization because it is not the topic of your work, then eliminate all the related information about urbanization from the text because you are confounding the lector about the topic of your work. 2) if on the contrary, you decide to keep going with the urbanization variable, please, include it correctly in your analysis either with a qualitative or quantitative variable, due to in the previous version of the manuscript authors claimed to already have this variable from a previous study. Besides, this will allow the authors clarify statements like did in lines 272-274 not depending in other studies, and if it is the case, they could discuss why they obtain different results for the same populations.

Apart from old issues, there are some new points than authors need to correct in the current version of the manuscript:

A) Authors propose in the introduction two main objectives (lines 70-72 and 78-81) that did not correspond with the ones analysed in the statistical analyses (line 165 and 173) and results. Please, reformulate correctly them.

B) Authors include references to supplementary material through all the results section, even though if they correspond to the principal results of their study. Thus, these parts should not be reported at supplementary but at principal tables. If the tables are too big, then prioritize the significant results or the most explicative and the others put it on supplementary material. The main results should be clarified along the manuscript with figures and tables and lectures have to get all the information with that, and only check supplementary material for extra information.

C) There is no Figure 3 in the new version of the manuscript, please be aware that all the information needed is included.

D) Significant interactions should be more clarified and explained in the results: e.g. lines 224-227, 229-231 and 249-251.

E) Fat and muscle score variables were not explained in the results, but they were statistical significant results with fat in juveniles. Why? Please, add and discuss them.

F) The text is referring to figures that do not have the correct information e.g. line 227, sex is not include in figure 3; line 244-245 juvenile size is not include in figure 2b.

G) During the discussion section, some references are missing (lines 285-286, 321-322) and others are included but not explained in line 268, 295, 297 and specially in 321.

H) The sentence in lines 161-162 should be included in results, not in materials and methods.

Reviewer #3: I have only a few minor suggestions that should be addressed in this manuscript before publication.

Line 51 – Places with birds presenting higher parasitemia had lower bird survival. Please correct this sentence because parasitemia per se is not the reason for the population decline.

Line 117 – keep this sentence in the past tense.

Line 121 – What does “experimenter” means? Do authors want to say something like “experienced researcher”? The same in line 129.

Line 180 – Change “injection” to “infection”.

Line 302 – Change to “thus are not sampled”.

7. PLOS authors have the option to publish the peer review history of their article (what does this mean?). If published, this will include your full peer review and any attached files.

Reviewer #1: No

Reviewer #2: No

Reviewer #3: Yes: Francisco Ferreira-Junior

---

## [Author Response · Author response to Decision Letter 1]

4 Jul 2020

PONE-D-20-04379R1

“Physiological and morphological correlates of blood parasite infection in urban and non-urban house sparrow populations”

- Our responses to comments are provided in bold and italic font. We also added a version of our manuscript where changes from the previous version appear highlighted in blue -

Dear Dr. Bichet,

Thank you for submitting your manuscript to PLOS ONE. After careful consideration, we feel that it has merit but does not fully meet PLOS ONE’s publication criteria as it currently stands. Therefore, we invite you to submit a revised version of the manuscript that addresses the points raised during the review process.

The reviewers and I came to the consensus that the manuscript still deserves attention. Two reviewers point out that the authors did not respond to the questions raised adequately. I strongly agree with them and emphasize that the authors must dedicate to answer the points raised with the utmost precision and to make all those reconsiderations in the manuscript to be submitted.

> We are very grateful to the associated editor and to the reviewers that our study still deserves attention. We are surprised to read that we did not answer some questions adequately. We are always happy and grateful to follow all the comments we obtained, since we really believe that the peer-review process is the best way to improve every studies. We did our best to include these suggestions. Some of them were very interesting and we addressed them. However, we are not convinced that others were relevant in the context of our study and we explained why in the previous round of review (it may be due to some misunderstanding and some repeated typos in the review of the reviewers, ex: reviewer 1 asked to correct for sample size, while he/she meant to correct by sample site, reviewer saying we did not take into account one previous request we cannot find in his/her first report). We now better understand what the reviewers meant and we have modified the manuscript according to most of the comments. For a few of them, we clearly justify why we have not processed with the suggested modifications (see our detailed answers below).

We would appreciate receiving your revised manuscript by Jun 20 2020 11:59PM. To enhance the reproducibility of your results, we recommend that if applicable you deposit your laboratory protocols in protocols.io, where a protocol can be assigned its own identifier (DOI) such that it can be cited independently in the future. For instructions see: http://journals.plos.org/plosone/s/submission-guidelines#loc-laboratory-protocols

• A rebuttal letter that responds to each point raised by the academic editor and reviewer(s). This letter should be uploaded as separate file and labeled 'Response to Reviewers'.

• A marked-up copy of your manuscript that highlights changes made to the original version. This file should be uploaded as separate file and labeled 'Revised Manuscript with Track Changes'.

• An unmarked version of your revised paper without tracked changes. This file should be uploaded as separate file and labeled 'Manuscript'.

We look forward to receiving your revised manuscript.

Kind regards,

Érika Martins Braga, Ph.D.

Academic Editor

PLOS ONE

Reviewers' comments:

Reviewer #1

In this new version, authors have been improved the contents of the manuscript, addressing some of the queries proposed by reviews, although I think most of the ‘content’ changes are still missing. The authors have addressed the majority of the queries proposed with a too vague and not entirely conclusive tone. Also, I have noticed that authors avoided to give concrete answers to important and fundamental questions from a data analysis point of view.

> We are very sorry we gave this impression to the reviewer. We are always happy and grateful to follow all the comments we obtained, since we really believe that the peer-review process is the best way to improve every studies. We did our best to include these suggestions. Some of them were very interesting and we addressed them. However, we are not convinced that others were relevant in the context of our study and we explained why in the previous round of review (it may be due to some misunderstanding and some repeated typos in the review of the reviewer, ex: the reviewer asked to correct for sample size, while he/she meant to correct by sample site). We now better understand what the reviewer meant and we have modified the manuscript according to most of the comments. For a few of them, we explain in details why we have not processed with the suggested modifications (see our detailed answers below).

The main problem I still see, is the lack of a complete model were all independent variables (physiological, morphological, and condition indices) are analysed at the same time by using infection status as dependent. Authors perform several little GLMs were they test different variables with infectious status but the overall effect is missing. Which are the most important variables? Which variable are driving infection in sparrows? I suggest this because the reader too often gets lost among so many non-significant results, tables, fragmented figures, supplementary information with significant results, sometimes some of them nor discussed in the manuscript.

I can suggest a general model with all the independent variables together in a AIC selection in order to have a more complete vision of the reality. 

> We are deeply sorry, but we have carefully read again the first round of reviews and we could not see where the reviewer asked for these full models in his/her first review. That is why we did not reply to this comment. We are also not convinced by this request because the physiological, morphological and condition variables were never the independent variables (see lines 180-184) in our models. Our goal was to understand whether these variables could be affected by blood parasite infections and not the opposite (all variables were entered as dependent variables, i.e. variables to be explained). We do not understand why the referee wants to have them as independent (explanatory) variables because it does not really make sense to test whether these variables could determine if a bird will be infected of not. The goal was to understand the impact of infection on organismal systems. 

Regarding the variables driving the infection, we have already conducted this model, including together all the relevant variables (population, capture date, age, sex, and now urbanisation score according to reviewer 1 and reviewer 2’s requests) which may affect the infection status (see lines 172-179). Therefore, we believe that we have already run the biologically-relevant full models to explain the infection status. We are sorry if we have missed something but we honestly do not think that the suggested models are relevant for this study.

However, to simplify our main message, we have shortened and clarified the result section (for instance lines 217, 231, 239, 244, 257, 240-242) and we have presented the results for adults and juveniles in the Tables 2 and 3 in the main manuscript and not anymore in the supplemental tables (Tables S1 and S2 deleted from the supplemental and added in Tables 2 and 3, respectively). But, to follow the reviewer 2’s request, we also have to provide more details regarding some significant interactions, in the result section, which lengthens this section a little bit (lines 231-235, 240-242, 250-258). 

In this model, authors should include a categorical variable of the sampling site (no size), as I previous commented them in the first review (but that authors said did not understand).

> We apologize we did not make this modification. In the first review, the reviewer requested to “try to control for the sampling size (N) as another covariate in your models. This could be a better option in order to control for the limitation of the small sampling size” and that “authors should control for the sampling site N in each population”. For us it was clear that the reviewer asked to control for sample SIZE and that the second comment contained a typo (site instead of size, because the “N” was mentioned in both, and N is usually used to refer to sample size). Regarding the current comment about sample SITE: all our models already include the sample site as a categorical variable since our first submission. This variable is called “population” (lines 177, 188) in our manuscript and all our tables (Tables 2 and 3). Therefore, we do not know how to satisfy the referee regarding this point: if the reviewer is talking about sampling SITE, this variable is already present in all our models. However, please note that according to next comments and reviewer 2’s request, we have modified our models and the population (or sampling site) is now added as a random factor in all our models (lines 177 and 188). 

Results about no differences between habitat category (urban/rural), should be also taken with care. As I suggest previously, habitat category (urban/rural) or sampling site (the four localities, as categorical), should have been a variable took into account in the other GLMs, and used as random factor (glmer function in R) when authors want to get pattern about independent variables that could be driving infection in birds, disengaging from environmental patterns.

> According to this comment, we have included the urbanization score as an explanatory variable and the sampling sites/populations as a random factor in our model (lines 113-114, 176 and 187, Table 1). This does not modify our results or our interpretations (Tables 2, 3, result and discussion sections).

Also, the manuscript has still lots of minor weaknesses that were pointed out in the text I attached.

> We thank the reviewer for the attached document and we have now corrected all the points mentioned in it (e.g. lines 41, 113, 157, 175, 196, 366).

Regarding the specific comment line 120: It is true that banding the bird is usually the first task after capture, but in this case, blood sampling has to be done within 3 minutes after capture to be able to access baseline corticosterone (lines 121-122), and therefore, blood sampling was definitely the first thing to do, before banding. 

Regarding the specific comment line 272, “Authors should include the previous ‘Urbanization Score’ they have in order to control for this effect and solve this problem”. As suggested, we have added these models in our manuscript (lines 112-113, 176 and 187) and we have modified the manuscript accordingly (methods, results and discussion). It does not change our results and there is no clear link between urbanization and the probability of infection (see also our answer to the previous related comment). 

Reviewer #2: 

Authors correctly clarify the most part of the issues raised in the first major revision, but they still have considerable points that were not solved in the first time, or needs to be clarify even more (previous point 3 to 5). In detail:

3) Authors should have standardized the use about “infection status or probability” terms. They have corrected it in the most part of the manuscript, but in the first part of results (lines 201-210), they still refer to “The number of infected individuals” which is not correct as in the table cited they use “infection status”. 

> We thank the reviewer for this suggestion and we now refer to “probability of being infected” (lines 211, 212, 214 and 216). We hope that this term will avoid any further confusion.

Moreover, in the figure 2B, they labelled the Y axis as “infection status” that is supposed to be a binary variable, but the axis has intermediate value. Why? Please, correct this ‘again’ and make sure that you standardize it all along the manuscript, including figures, tables, and Supplementary Information.

> We added the intermediate values to help to visualize predictions obtained from the model. For example, if you want to know the probability of infection for an adult bird caught at day 210, the model gives you a probability of infection around 0.5. We thought it was useful and informative, but we now deleted it and only keep the ticks 1 and 0 in the Figure 2B. We also corrected the legend of the Table 2 by replacing “blood parasite infection” by “infection status” (lines 749, 753, 764). 

4) Authors were asked to consider the results from Jiménez-Peñuela et al. 2019 in their study and even they include it in line 62 and 326, they did not consider it in their third hypothesis proposed in the discussion, that is the same hypothesis proposed in this work as I said in the first review. Moreover, in lines 66-68 they made an inappropriate statement “has never been examined in house sparrow”, but during all the introduction they are referring to works that already have evaluated similar variables in house sparrows, including in the suggested reference. The same occur with the statement on lines 277-278, where I suggested softening the tone in minor comments. Please, correct any similar statements along all the manuscript ‘again’ and make sure to check what other authors have recently did in the same topic, especially the ones that you are referring along your work.

> We apologize for this mistake. We have now included this reference when explaining the third hypothesis of the discussion (line 362). We have also removed the terminology “has never been examined in house sparrows” from the manuscript and we have made sure to refer to previous works (lines 67-68).

5) Authors were asked to include either a qualitative or quantitative variable evaluating the urbanization of their population, due to as I said, including population ID in the statistical analysis only indicates if there was any difference between populations, but it did not compare urban and non-urban ones. Authors are claiming through all the manuscript that they compare urban and non-urban population (title, keywords, introduction, discussion…) but their answer to this request was: “The influence of urbanization on physiological parameter is not the topic of our study and we added population ID to control for population differences in physiological and morphological parameters already found in the previous study (Meilliere et al 2015 PlosOne 10(8))”.

I don't entirely agree with that answer and I can only comprehend two options: 1) if you do not include any variable that evaluate urbanization because it is not the topic of your work, then eliminate all the related information about urbanization from the text because you are confounding the lector about the topic of your work. 2) if on the contrary, you decide to keep going with the urbanization variable, please, include it correctly in your analysis either with a qualitative or quantitative variable, due to in the previous version of the manuscript authors claimed to already have this variable from a previous study. Besides, this will allow the authors clarify statements like did in lines 272-274 not depending in other studies, and if it is the case, they could discuss why they obtain different results for the same populations.

> We agree and we have now added the urbanization score as a quantitative variable to take into consideration the level of urbanization (urbanisation score, lines 113-114) on our data (lines 176 and 187). We also now explicitly mention that the impact of urbanization on these variables has been previously published with the same data set and that we specifically look at the effect of malaria infection on these variables in different populations (along an urbanization gradient) (lines 292-295 and 324-327).

Apart from old issues, there are some new points than authors need to correct in the current version of the manuscript:

A) Authors propose in the introduction two main objectives (lines 70-72 and 78-81) that did not correspond with the ones analysed in the statistical analyses (line 165 and 173) and results. Please, reformulate correctly them.

> We have modified accordingly lines 72-74 and 81.

B) Authors include references to supplementary material through all the results section, even though if they correspond to the principal results of their study. Thus, these parts should not be reported at supplementary but at principal tables. If the tables are too big, then prioritize the significant results or the most explicative and the others put it on supplementary material. The main results should be clarified along the manuscript with figures and tables and lectures have to get all the information with that, and only check supplementary material for extra information.

> Accordingly, we have shortened and restructured the results section to be clearer (for instance lines 217, 231, 239, 244, 257, 240-242). Moreover, we now present the results for adults and juveniles in the Tables 2 and 3 in the main manuscript and not anymore in the supplemental (Tables S1 and S2 deleted from the supplemental and added in Tables 2 and 3, respectively). We hope that this will satisfactorily answer to the reviewer’s comment.

C) There is no Figure 3 in the new version of the manuscript, please be aware that all the information needed is included.

> We apologize for this mistake. Strangely, the Figure 3 was included in the pdf we received right after our resubmission. We definitely had a Figure 3 and we are making sure it is included in this new submission. 

D) Significant interactions should be more clarified and explained in the results: e.g. lines 224-227, 229-231 and 249-251.

> According to this comment, we have modified the results to explain better the most relevant significant interactions (lines 231-235, 240-242 and 250-258).

E) Fat and muscle score variables were not explained in the results, but they were statistical significant results with fat in juveniles. Why? Please, add and discuss them.

> Fat and muscle scores were both not correlated with infection status in either adults or juveniles (see Table 3C), that is why there are not discussed (the aim of the analysis was to test if infection status could explain fat/muscle scores). We only mentioned that “Infection status was not associated with sparrow condition (body condition, fat, and muscle scores, Table 3c)” (lines 248-259). The fact that fat score differed between juveniles and adults (Table 3C) is not the purpose of the present study focusing on blood parasite infections and physiological, morphological and condition consequences. In these models, “age” was added as a covariate to control for expected physiological, morphological and condition differences between adults and juveniles. We have added a sentence (lines 324-327) stating that these differences were already found in a previous study using the same data set (and aiming to understand the effect of urbanization on these parameters).

F) The text is referring to figures that do not have the correct information e.g. line 227, sex is not include in figure 3; line 244-245 juvenile size is not include in figure 2b.

> Thank you. This is corrected (lines 235, 245).

G) During the discussion section, some references are missing (lines 285-286, 321-322) and others are included but not explained in line 268, 295, 297 and specially in 321.

> We included references accordingly (lines 286, 335), in a limited number, to still fit with the first round of comments stating that we had too many references, and we explained the references (lines 267-268, 298-300, 302, 330-334). 

H) The sentence in lines 161-162 should be included in results, not in materials and methods.

> We understand why the reviewer would like this sentence to be moved to the results. We would prefer to keep it here, otherwise it will make some confusion about which parasite is taking into account to determine the infection status used in all the analyses. If the reviewer still thinks that it should be moved to the results after reading these explanations, we will be happy to do it.

 

Reviewer #3: 

I have only a few minor suggestions that should be addressed in this manuscript before publication.

Line 51 – Places with birds presenting higher parasitemia had lower bird survival. Please correct this sentence because parasitemia per se is not the reason for the population decline.

> Thank you. This is now corrected (line 51). 

Line 117 – keep this sentence in the past tense.

> Corrected (line 122). 

Line 121 – What does “experimenter” means? Do authors want to say something like “experienced researcher”? The same in line 129.

> “Experimenter” means the person who made the measurements. Nothing related with the level of experience. We checked with a native speaker and this term seems appropriate in our respect. 

Line 180 – Change “injection” to “infection”.

> Modified (line 189). 

Line 302 – Change to “thus are not sampled”. 

> Modified (line 308).

---

## [Decision Letter · Decision Letter 2]

17 Jul 2020

PONE-D-20-04379R2

Physiological and morphological correlates of blood parasite infection in urban and non-urban house sparrow populations

PLOS ONE

Dear Dr. Bichet,

Thank you for submitting your manuscript to PLOS ONE. After careful consideration, we feel that it has merit but does not fully meet PLOS ONE’s publication criteria as it currently stands. Therefore, we invite you to submit a revised version of the manuscript that addresses the points raised during the review process.

The authors addressed the main concerns from the *reviews*. However, your revised manuscript still deserves attention. Please, provide, point-to-point responses according to the comments made the Reviewer 21 in the new version of your manuscript. 

We look forward to receiving your revised manuscript.

Kind regards,

Érika Martins Braga, Ph.D.

Academic Editor

PLOS ONE

Reviewers' comments:

Reviewer's Responses to Questions

**Comments to the Author**

1. If the authors have adequately addressed your comments raised in a previous round of review and you feel that this manuscript is now acceptable for publication, you may indicate that here to bypass the “Comments to the Author” section, enter your conflict of interest statement in the “Confidential to Editor” section, and submit your "Accept" recommendation.

Reviewer #1: All comments have been addressed

Reviewer #2: (No Response)

2. Is the manuscript technically sound, and do the data support the conclusions?

Reviewer #1: Yes

Reviewer #2: Yes

3. Has the statistical analysis been performed appropriately and rigorously? 

Reviewer #1: Yes

Reviewer #2: Yes

4. Have the authors made all data underlying the findings in their manuscript fully available?

Reviewer #1: Yes

Reviewer #2: Yes

5. Is the manuscript presented in an intelligible fashion and written in standard English?

Reviewer #1: Yes

Reviewer #2: Yes

6. Review Comments to the Author

Reviewer #1: In this new version, authors have been considerably improved the contents of the manuscript, addressing almost all the queries proposed by reviewers. The statistical analyses have improve considerably. Though, the manuscript has few minor faults that were pointed out in the text I attached.

Reviewer #2: Authors clarify correctly and solve the issues presented in the first and second major revisions. Even though the main limitations of the study that was the number of individuals use and genera identity of the parasite is still present due to it is not possible to be solved, the manuscript has improved. I would recommend assessing some small points:

Line 122: “The blood collected was also to be used”?? maybe they should delete “to be”.

Lines 196-197: Adult and juvenile information is no longer in supplementary information. They should correct this reference.

Lines 203-204: If they say that interactions with a p-value higher than 0.05 were removed from all the models, why in Table 3 B with the analyses of body mass in all birds they keep the interaction infection status-age, infection status-sex and sex-age? Please correct this.

Lines 233 and 242: Which are these post-hoc comparison? Please, specify the function used in the statistical analyses.

7. PLOS authors have the option to publish the peer review history of their article (what does this mean?). If published, this will include your full peer review and any attached files.

Reviewer #1: No

Reviewer #2: No

---

## [Author Response · Author response to Decision Letter 2]

18 Jul 2020

PONE-D-20-04379R2

“Physiological and morphological correlates of blood parasite infection in urban and non-urban house sparrow populations”

- Our responses to comments are provided in bold and italic font. We also added a version of our manuscript where changes from the previous version appear highlighted in blue -

Dear Dr. Bichet,

Thank you for submitting your manuscript to PLOS ONE. After careful consideration, we feel that it has merit but does not fully meet PLOS ONE’s publication criteria as it currently stands. Therefore, we invite you to submit a revised version of the manuscript that addresses the points raised during the review process.

The authors addressed the main concerns from the reviews. However, your revised manuscript still deserves attention. Please, provide, point-to-point responses according to the comments made the Reviewer 21 in the new version of your manuscript. 

> We would like to thank the associate editor and the two reviewers for their comments. We replied point by point to them and we hope that this revised version will provide entire satisfaction. 

We look forward to receiving your revised manuscript.

Kind regards,

Érika Martins Braga, Ph.D.

Academic Editor

PLOS ONE

Reviewers' comments:

Reviewer #1: In this new version, authors have been considerably improved the contents of the manuscript, addressing almost all the queries proposed by reviewers. The statistical analyses have improve considerably. Though, the manuscript has few minor faults that were pointed out in the text I attached.

> We would like to thank the reviewer for saying that our manuscript is considerably improved. We also thank the reviewer for taking the time to edit our manuscript to correct the minor remaining mistakes we had. We corrected that (e.g. lines 49, 60, 87, 274, 280, 308, 430, 608, 725). We also corrected the Tables 2 and 3 by giving three decimals to all values. 

Reviewer #2: Authors clarify correctly and solve the issues presented in the first and second major revisions. Even though the main limitations of the study that was the number of individuals use and genera identity of the parasite is still present due to it is not possible to be solved, the manuscript has improved. 

> We would like to thank the reviewer for thinking the manuscript has improved and that we clarified and solved the issues she/he pointed out. 

I would recommend assessing some small points:

Line 122: “The blood collected was also to be used”?? maybe they should delete “to be”.

> Deleted (line 123). 

Lines 196-197: Adult and juvenile information is no longer in supplementary information. They should correct this reference.

> Thank you. It is corrected (line 196). 

Lines 203-204: If they say that interactions with a p-value higher than 0.05 were removed from all the models, why in Table 3 B with the analyses of body mass in all birds they keep the interaction infection status-age, infection status-sex and sex-age? Please correct this.

> We apologize, but we cannot access this request. If the two-way interactions are not significant, the three-way interaction was significant, and therefore, the associated two-way interaction as well as the single terms have to remain in the model. 

Lines 233 and 242: Which are these post-hoc comparison? Please, specify the function used in the statistical analyses.

> We added this information (lines 204-206).

---

## [Editor Report · Decision Letter 3]

22 Jul 2020

Physiological and morphological correlates of blood parasite infection in urban and non-urban house sparrow populations

PONE-D-20-04379R3

Dear Dr. Bichet,

We’re pleased to inform you that your manuscript has been judged scientifically suitable for publication and will be formally accepted for publication once it meets all outstanding technical requirements.

Kind regards,

Érika Martins Braga, Ph.D.

Academic Editor

PLOS ONE
---

## [Editor Report · Acceptance letter]

5 Aug 2020

PONE-D-20-04379R3 

Physiological and morphological correlates of blood parasite infection in urban and non-urban house sparrow populations 

Dear Dr. Bichet:

I'm pleased to inform you that your manuscript has been deemed suitable for publication in PLOS ONE. Congratulations! Your manuscript is now with our production department. 

Kind regards, 

on behalf of

Dr. Érika Martins Braga 

Academic Editor

PLOS ONE